



# Importance of high resolution nitrogen deposition data for biogeochemical modeling in the western Baltic Sea and the contribution of the shipping sector

Daniel Neumann[1], René Friedland[1], Matthias Karl[2], Hagen Radtke[1], Volker Matthias[2], and Thomas Neumann[1]

[1]Leibniz-Institute for Baltic Sea Research Warnemünde, Seestr. 15, 18119 Rostock, Germany
[2]Institute of Coastal Research, Helmholtz-Zentrum Geesthacht, Max-Planck-Str. 1, 21502 Geesthacht, Germany

**Correspondence:** Daniel Neumann (daniel.neumann@io-warnemuende.de)

**Abstract.** Atmospheric deposition accounts for up to a third of the nitrogen input into the Baltic Sea and contributes to eutrophication. It is useful to use three-dimensional biogeochemical models to evaluate the contribution of atmospheric nitrogen deposition to eutrophication because bioavailable nitrogen impacts eutrophication differently depending on time and place of input – e.g. nitrogen is processed and denitrified faster in flat coastal regions. The western Baltic Sea, which is stressed by high nutrient loads, is characterized by many small islands and a wrinkled coast line. In regions with this type of coastal features, the grid resolution of atmospheric chemistry transport models (CTMs) has a strong impact on the modeled nitrogen deposition. The aim of this study was to evaluate the benefit of finer spatially resolved deposition data as input for simulations with the ecosystem model ERGOM. This study also focused on the shipping contribution to the marine nitrogen budget via deposition of shipping-emitted nitrogen oxide ($NO_X$). Differences in the modeled dissolved inorganic nitrogen (DIN) caused by refined nitrogen deposition were identified in some coastal sections and between the Danish islands. Patches of enhanced DIN concentrations were found distant to the coast in model runs forced by the finer resolved data. These were caused by better resolved precipitation events. The differences between fine and coarse resolution deposition of the same CTM were low compared to the difference to EMEP deposition, which was a third comparison data set. The shipping sector contributed a maximum of $10\%$ and on average less than $5\%$ to DIN. In summary, particularly small scale ecosystem model studies in bights are expected to benefit from spatially higher resolved nitrogen deposition data. The shipping sector is a relevant contributor to the marine nitrogen deposition but its contribution to the marine DIN pool is rather low.

*Copyright statement.* TEXT

# 1 Introduction

Human activities have had a considerable impact on the Baltic Sea for centuries. Its ecosystem is exposed to growing economic and anthropogenic pressures (Andersen et al., 2015; Korpinen et al., 2012; Svendsen et al., 2015). One of these pressure



factors is the excessive input of nutrients, i.e. bioavailable nitrogen and phosphorus compounds, which leads to eutrophication (Svendsen et al., 2015). Although the eutrophication status has improved over the past three decades (Andersen et al., 2017; Svendsen et al., 2015; Gustafsson et al., 2012), a Good Environmental Status (GES) is not restored yet (e.g., HELCOM, 2009). Therefore, the descriptor 5 of the Marine Strategy Framework Directive (MSFD) and the Baltic Sea Action Plan (BSAP) still

focus on eutrophication (EU-2008/56/EC, 2008; HELCOM, 2007).

Riverine nutrient loads have been evaluated in detail in the past decades (Sutton et al., 2011; Nausch et al., 2017; StÅlnacke et al., 1999; HELCOM, 2013a; Svendsen et al., 2015) and agricultural activities being identified as the dominant source of high nutrient loads in recent years (Andersen et al., 2016). River inflow accounts for approximately $2/3$ to $3/4$ of the bioavailable nitrogen input and atmospheric deposition approximately for $1/4$ to $1/3$ (HELCOM, 2013a, b). Therefore, atmospheric deposi-

tion is not negligible in the context of eutrophication (Simpson, 2011; HELCOM, 2005; Svendsen et al., 2015). Although data products of spatio-temporally resolved oxidized and reduced nitrogen deposition exist (EMEP, 2017), studies on the contribution of different atmospheric emission sectors to nitrogen deposition are rare and dominantly focused on the shipping sector (Bartnicki and Fagerli, 2008; Tsyro and Berge, 1998; HELCOM, 2005; Stipa et al., 2007a; Bartnicki et al., 2011; Hongisto, 2014). The knowledge on the spatio-temporal distribution of atmospheric phosphate deposition is low – particularly due to

a lack of detailed phosphate emission data for model studies (Gustafsson et al., 2012; HELCOM, 2013a; Rolff et al., 2008; HELCOM, 2015; Mahowald et al., 2008; Tipping et al., 2014). Currently, a constant phosphate deposition of $5\ \mathrm{mg\ m^{-2}\ a^{-1}}$ is assumed for the Baltic Sea (HELCOM, 2013b). A revision and improvement of this figure is ongoing (HELCOM, 2017).

Atmospheric nitrogen deposition is higher above land than above water because of a higher surface roughness leads to a higher dry deposition velocity (Seinfeld and Pandis, 2016, Chap. 19). Nevertheless, coastal waters are considerably impacted

by atmospheric nitrogen deposition: the largest atmospheric emission sources of oxidized and reduced nitrogen compounds are located on the land (CEIP, 2018) and coastal waters are closer to these sources than open ocean waters. Additionally, some gaseous nitrogen compounds condense on coarse sea salt particles, which have a short atmospheric residence time, and, hence, deposit faster into the ocean (Paerl et al., 2002; Neumann et al., 2016).

When nitrogen deposition is calculated by atmospheric chemistry transport models (CTMs), grid cell boundaries of the

models do not necessarily follow the coastline. Therefore, coastal grid cells often contain a mixture of nitrogen deposition onto land and into water, which sum is written out as one value per grid cell. The constant area-specific deposition all over the grid cell leads to overestimation of marine nitrogen deposition in coastal grid cells when it is extracted from CTM output files. In contrast, deposition onto land is underestimated in these grid cells. Therefore, an adaption of the model grid to the coastline or refining the grid resolution can be expected to improve the modeled nitrogen deposition. Estimating marine coastal nitrogen

deposition based on measurements involves similar problems (Meyers et al., 2013). The western Baltic Sea is a region with a highly structured coastline and includes several islands. Thus, it an example for a region, which is probably affected by the described nitrogen deposition issues.

In coastal regions, marine biogeochemical models which consider atmospheric deposition might benefit from higher resolved atmospheric models and a more accurate nitrogen deposition. Further, finer resolved atmospheric models – including meteorology –

are more likely to reproduce small-scale precipitation events, which are explicitly modeled on finer resolutions



instead of being parameterized. Since precipitation is the main driver of wet deposition, the quality of nitrogen deposition predictions is likely to be improved. This affects not only coastal but also open ocean regions.

The shipping sector is an important contributor to atmospheric nitrogen oxide ($NO_X$) air pollution in Europe and also in the Baltic Sea Region (Jonson et al., 2015; Aksoyoglu et al., 2016). Thus, it considerably contributes to nitrogen deposition

– particularly at the open sea. Tsyro and Berge (1998) found that the shipping sector contributed $5\%$ to $10\%$ to the $NO_X$ deposition in the Baltic Sea in 1990. The shipping sector contributed approximately $6\%$ to the total nitrogen deposition in 2000 (HELCOM, 2005) and approximately $14\%$ to the oxidized nitrogen deposition in 2005 (Bartnicki and Fagerli, 2008). In 2010 approximately $13,500\ \mathrm{t/a}$ and $9,500\ \mathrm{t/a}$ of the nitrogen deposition into the Baltic Sea originated from Baltic Sea and North Sea shipping, respectively. The total atmospheric nitrogen deposition accounted for $218,600\ \mathrm{t/a}$ and the waterborne

nitrogen input for $758,300\ \mathrm{t/a}$ (HELCOM, 2013b). A specific target for a reduction of the annual nitrogen deposition from shipping was set to $5,735\ \mathrm{t/a}$ (HELCOM, 2013c) within the latest revision of the HELCOM Baltic Sea Action Plan.

The North Sea and Baltic Sea will be declared as Nitrogen Emission Control Areas (NECAs) according to MARPOL Annex VI from 2021 onwards. This means that oceangoing ships, which are built after 2021, have to comply with "Tier III emission thresholds" when they enter the North Sea and Baltic Sea regions. These Tier III emission thresholds force emission reductions

of $75\%$ to $80\%$ compared to the currently valid Tier I and Tier II thresholds. Hence, $NO_X$ emissions of individual ships are expected to decline from 2021 onwards. However, shipping traffic is also expected to increase in the Baltic Sea in the next decades (e.g., Matthias et al., 2016; Karl et al., in prep. same spec. issue a).

The impact of one input source sector, i.e. shipping, on the marine biogeochemistry does not only depend on its annual input but also on the residence times of nutrients in the system. These residence times are also governed by the location and time of

the nutrient release as well as by the availability of other nutrients. Hence, the amount of shipping-related nitrogen deposition relative to other nitrogen inputs is not necessarily linearly related to its impact. Using a nutrient source tagging approach (e.g., Ménesguen et al., 2006) one is able to trace the contribution of the shipping sector in biogeochemical key variables in order to evaluate the impact of shipping emissions on biogeochemical processes.

We derived two research topics for this study based on the current state of knowledge:

1. What resolution of atmospheric nitrogen deposition is necessary to properly evaluate the impact of nitrogen deposition on DIN concentrations and biomass production in highly structured coastal areas?

2. How high is the contribution of shipping related nitrogen deposition to this?

For topic one the expected benefits of using higher resolved nitrogen deposition data in the German and Danish Baltic Sea waters with respect to modeling biogeochemical key variables is evaluated. Two nitrogen deposition data sets, which were

calculated with the same CTM but with different spatial resolution and different precipitation parameterizations, are used for this purpose. Additionally, we use EMEP nitrogen deposition data, which is used for the official HELCOM reporting, to get a feeling for inter-model variability.

For topic two, we used an established source tagging approach (Ménesguen et al., 2006; Neumann, 2007; Radtke et al., 2012) to trace the contribution of shipping-related nitrogen in chosen biogeochemical key variables.



**Table 1.** Acronyms and sources of nitrogen deposition data sets used in this study.

| Acronym | CTM | Resolution | Meteo | Meteo-Resol. | Ship emission inventory |
|---------|-----|-----------|-------|--------------|------------------------|
| cmaq16 | CMAQ v5.0.1, cb05tucl, aero5 | $16 \times 16 \text{ km}^2$ | CCLM v5.0, spectral nudging with ERA-Interim reanalysis | $0.11° \times 0.11°$ | individual shipping contribution based on AIS ship positions |
| cmaq04 | CMAQ v5.0.1, cb05tucl, aero5 | $4 \times 4 \text{ km}^2$ | CCLM v5.0, spectral nudging with ERA-Interim reanalysis; precip.: convection-permitting | $0.025° \times 0.025°$ | individual shipping contribution based on AIS ship positions |
| emep | EMEP rv4.9, 2016 reporting | $50 \times 50 \text{ km}^2$ | IFS-ECMWF, v40r | T1279 (16 km) | Ship emission inventory from TNO MAAC-III |

To the best of our knowledge, this is the first study, in which nutrient tagging in biogeochemical models is applied to specific atmospheric deposition sectors for the Baltic Sea. Previously, it has been applied to riverine inflow (Radtke et al., 2012) and salt water inflow events (Neumann et al., 2017). Raudsepp et al. (2013) performed a similar study focusing on the impact of shipping-related nitrogen deposition on nitrogen fixation by cyanobacteria in the Gulf of Finland. However, they did not

tag shipping-related nitrogen but performed two simulations: one with and another without shipping nitrogen contribution and calculated the difference. Tagging of atmospheric nitrogen deposition has been done for the North Sea and the English Channel in a few studies (Große et al., 2017; Los et al., 2014; Troost et al., 2013; Ménesguen et al., 2018; Dulière et al., 2017). The method has also been used to tag nitrogen compounds in atmospheric chemistry transport model simulations (e.g., Brandt et al., 2011; Geels et al., 2012; Wu et al., 2011).

**2 Materials and Methods**

Three nitrogen deposition data sets were used as input data for the biogeochemical model HBM-ERGOM. Two data sets were simulation results of the Community Multiscale Air Quality (CMAQ) atmospheric chemistry transport model: one with $16 \times 16 \text{ km}^2$ horizontal resolution and one with $4 \times 4 \text{ km}^2$ denoted as cmaq16 and cmaq04. Based on these two data sets, the evaluation of refining the model grid resolution is performed. The third nitrogen deposition data set was taken from the

European Measurement and Evaluation Programme (EMEP), which is used for the official HELCOM nitrogen deposition reporting and has a spatial resolution of $50 \times 50 \text{ km}^2$. This data set from another CTM was chosen to analyze the impact of a refined model resolution compared to the variability between different models. Because the EMEP data is much coarser resolved than the CMAQ data, we will see a combined impact of a different model and different resolution.

The cmaq04 and cmaq16 deposition cases contain total and shipping related nitrogen deposition. Therefore, both deposition

data sets could be used for an assessment of the shipping impact on the biogeochemistry of the Baltic Sea.

The model setups are described below and briefly summarized in Table 1.



## 2.1 Atmospheric Modeling

The CMAQ model is maintained and provided by the U.S. Environmental Protection Agency (US EPA). CMAQ version 5.0.1 (Nolte et al., 2015; Foley et al., 2010; Appel et al., 2017) with the cb05tucl gas phase chemistry mechanism (Sarwar et al., 2007; Whitten et al., 2010; Yarwood et al., 2005) and aero5 aerosol chemistry, which is based on ISORROPIA v1.7 (Fountoukis and

Nenes, 2007; Sarwar et al., 2011), was used for this study. Atmospheric particles are represented by a three-moment schema containing three size modes (Binkowski and Roselle, 2003). The dry deposition parameterization for particulate matter is an updated version of Binkowski and Shankar (1995), which is based on Slinn and Slinn (1980) and Pleim et al. (1984). The parameterization considers gravitational settling, aerodynamic resistance above the canopy, and surface resistance. The three modes and the three moments are deposed individually. Land based emissions were aggregated with SMOKE for Europe

(Bieser et al., 2011). Marine shipping emissions were calculated according to Aulinger et al. (2016) for the North Sea and with the STEAM model (Jalkanen et al., 2012) for the Baltic Sea. Sea salt emissions were calculated inline (Gong, 2003; Kelly et al., 2010) without surf zone emissions (Neumann et al., 2016).

The CMAQ simulations were performed on three one-way nested model domains with increasing horizontal grid resolution (Fig. 1) and each 30 vertical z-layers. The outer model domain ($64 \times 64$ km$^2$ grid resolution) covered Europe and northern

Africa and the lateral boundary conditions were taken from FMI APTA global reanalysis (Sofiev et al., 2018). The first nested model domain ($16 \times 16$ km$^2$ grid resolution) covered the North Sea and Baltic Sea regions. Into this domain, a finer resolved model grid ($4 \times 4$ km$^2$) was nested, which covered the western Baltic Sea. The model runs are denoted as cmaq64, cmaq16, and cmaq04, respectively. Nitrogen deposition data from the cmaq16 and cmaq04 simulations were used as atmospheric input data for the biogeochemical modeling experiments.

Meteorological input data for the CMAQ simulations were modeled with COSMO-CLM (CCLM) version 4.8_clm_11 with spectral nudging (Rockel et al., 2008). The cmaq64 and cmaq16 runs were forced by a CCLM simulation on a rotated grid of $0.11°$ spatial resolution (rotated North Pole located at $170°$ W, $35°$ N). This data set is available as coastDat2 atmosphere dataset (Geyer and Rockel, 2013; Geyer, 2014). The cmaq04 run was forced by a CCLM simulation of $0.025°$ spatial resolution with a higher degree of resolved precipitation.

Karl et al. (in prep. same spec. issue a) describes the model setup in more detail and Karl et al. (in prep. same spec. issue b) presents a validation for the cmaq04 simulation results.

As a third nitrogen deposition data set, the EMEP deposition data of the 2016 reporting were used (EMEP, 2016). These data were calculated by the EMEP MSC-W model revision 4.9 of the Norwegian Institute for Air Research (NILU) on the default $50 \times 50$ km$^2$ EMEP model domain (Simpson et al., 2012) and downloaded from the EMEP homepage in daily resolution

(EMEP, 2018). The EMEP model was driven by meteorological data from the Integrated Forecasting System (IFS) of the European Centre for Medium-Range Forecast (ECMWF). The emissions were provided by the Center for Emission Inventories and Projections (CEIP) except for shipping emissions, which were taken from the TNO-MACC-III emission inventory of 2011 (EMEP, 2015; Kuenen et al., 2014; MACC-III, 2016). The latter emissions are not based on AIS data as the emissions for the CMAQ simulations do. Total annual emission data of the CEIP emission inventory were also used to calculate the CMAQ





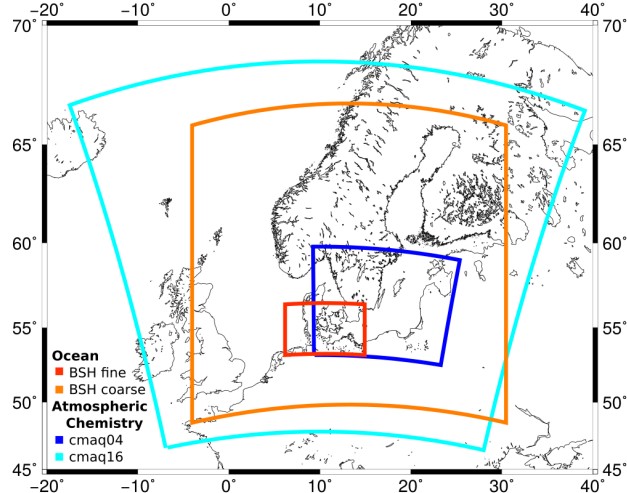

**Figure 1.** Extent of the model domains of the atmospheric chemistry transport and the marine biogeochemical models

model land-based emissions with SMOKE for Europe (Bieser et al., 2011). The original EMEP land-sea mask has a resolution of $5 \times 5$ km$^2$. Because the EMEP model simulation are performed and written out on $50 \times 50$ km$^2$ grid resolution, it is not possible to obtain a nitrogen deposition data set finer resolved than $50 \times 50$ km$^2$. A recent comparison of EMEP model results with results of other air quality models and with measurements is documented in Bian et al. (2017).

CMAQ and the EMEP model differ with respect to some important processes. Hence, differences in the nitrogen deposition data sets cannot be attributed to solely one factor. The formation of ammonium nitrate in EMEP is different than in CMAQ, because EMEP considers a higher rate for the heterogeneous reaction of $N_2O_5 + H_2O$ leading to nitrate. The parameterization of dry deposition to sea is also different. Detailed information on the processes and parameterizations in the EMEP model are provided in the recent EMEP reports (EMEP, 2015, 2016) and in Simpson et al. (2012).

All nitrogen deposition data sets were bilinearly interpolated onto the HBM-ERGOM model grid resolution with Climate Data Operators (cdo) v1.7.0 (CDO, 2018) and supplied as daily mean values.

## 2.2    Marine Modeling

The biogeochemical modeling was performed with the Ecological ReGional Ocean Model (ERGOM, https://ergom.net/). It was online-coupled to the physical model HBM (HIROMB-BOOS-Model). HBM evolved from the BSH circulation model
(BSHcmod) (Dick and Kleine, 2007) and is maintained by the Danish Meteorological Institute (DMI) and the Federal Maritime and Hydrographic Agency of Germany (BSH) (Brüning et al., 2014; Poulsen et al., 2015). HBM is typically used for operational forecasts, but was run in hindcast mode for this study.

ERGOM was originally developed to represent biogeochemical processes in the Baltic Sea (Neumann, 2000; Neumann et al., 2002). It has been extensively used, validated (e.g., Eilola et al., 2011), and extended in the past 15 years leading to different



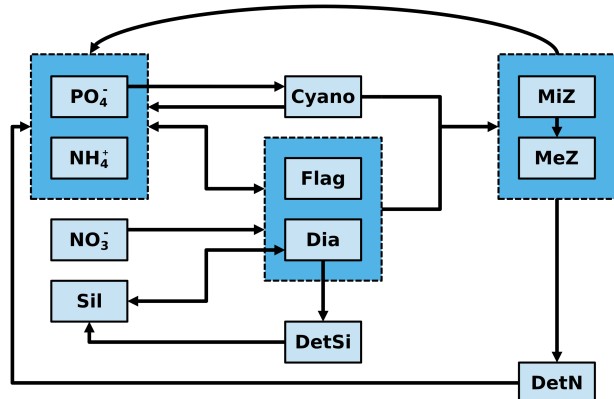

**Figure 2.** Schematic overview of state variables and their relations in ERGOM used in this study. Two tracers – oxygen and liable dissolved organic nitrogen – are not shown to simplify the diagram.

branches of the original version (e.g., Kuznetsov et al., 2008; Lessin et al., 2014; Miladinova and Stips, 2010; Neumann et al., 2015; Radtke et al., 2012; Neumann and Schernewski, 2005; Friedland et al., 2012; Schernewski and Neumann, 2005; Schernewski et al., 2015). The version which is employed in this study has been adapted for modeling a coupled North Sea and Baltic Sea ecosystem (Maar et al., 2011). Therefore, silicate was included as a limiting nutrient, which is important for the

North Sea biogeochemistry, e.g. (Reid1990).

The ERGOM in this study consists of 13 tracers. Figure 2 shows eleven of these tracers and their relations. Oxygen and labile dissolved organic nitrogen are not included in order to simplify the structure of the diagram. Phosphate ($PO_4^-$), ammonium ($NH_4^+$), nitrate ($NO_3^-$), and silicate (Sil, $SiO_4^-$) are nutrients for large phytoplankton (diatoms, dia), small phytoplankton (flagellates, flag), and cyanobacteria. The latter are able to fix $N_2$ instead of using nitrate and ammonium. The later three

tracers are grazed by microzooplankton and mesozooplankton. Silicate detritus (DetSi) is released when diatoms are consumed. Zooplankton respirates phosphate and ammonium and is converted to detritus (DetN) at its end of life. Detritus is converted to the original nutrients. Documentation of model constants and processes is provided in the supplement to this publication.

The ERGOM model was extended by a method for nutrient tagging, which is described by Ménesguen et al. (2006) and implemented by Neumann (2007) and Radtke et al. (2012). This method enables the modeler to tag elements, e.g. nitrogen,

from a specific source, e.g. shipping, and track it through all model state variables. Thus, it is possible to trace the contribution of individual sources through the ecosystem without disturbing the system dynamics, as deactivating specific sources would do. In this specific case, nitrogen deposition from shipping emissions was tagged in ERGOM.

HBM-ERGOM was run on two nested domains (Fig. 1). The outer domain ($5' \times 3'$ horizontal grid resolution $\approx 5 \times 5$ km$^2$, 36 vertical z-star layers) covers the whole North Sea and Baltic Sea. The second domain ($50'' \times 30''$ horizontal grid resolution

$\approx 0.9 \times 0.9$ km$^2$, 25 vertical z-star layers) is two-way nested into the outer one and covers German territorial waters and the Kattegat.





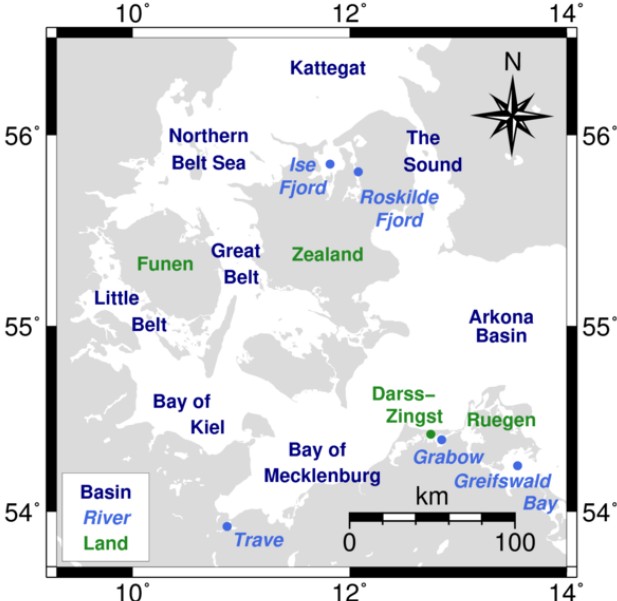

**Figure 3.** Geographic locations mentioned in this publication. Basins of the Baltic Sea and of the Kattegat are printed in navy blue. Locations of rivers, Fjords, and Lagoons are printed in light blue. Islands and peninsulas are printed in green. Some locations are marked with dots to prevent ambiguity in the naming.

Three HBM-ERGOM simulations were performed with the three nitrogen deposition data sets cmaq04, cmaq16, and EMEP. Other parameters were kept unchanged. Due to model limitations, daily mean deposition data was provided for the HBM-ERGOM model runs and not hourly as would be available from CMAQ. The phosphate deposition was assumed to be constant $(0.471 \, \mu\text{mol m}^{-2} \, \text{d}^{-1} \approx 5.33 \, \text{mg m}^{-2} \, \text{a}^{-1})$ . Except for $O_2$, which is exchanged between atmosphere and dissolved phase

based on its partial pressure difference, no air-sea exchange of gases or particles was taken into account. We denote the three HBM-ERGOM model runs, which were driven by the three deposition data sets, as cmaq04, cmaq16, and EMEP "model cases".

HBM-ERGOM was run for two years for each deposition case. Nutrient tagging of shipping-related nitrogen was activated in the cmaq04 and cmaq16 cases. The first year was considered as spin-up and only the second year was evaluated for this study.

The initial conditions for the first year were taken from a pre-existing HBM-ERGOM model run with constant atmospheric deposition and without nutrient tagging.

## 2.3 Study region

The western Baltic Sea was choosen as study region. It is bordered by land in the south, west, and northeast. Danish islands like Zealand and Funen are located in the center of this region. Islands and water bodies, which are mentioned in this manuscript

are labeled in Fig. 3.



**Table 2.** Measurement stations, which are presented for model evaluation in Sect. 3. Coordinates and approximate geographic location of each station are provided in the second to fourth columns. The last column lists the sources of measurement data.

| Station Name | Lon | Lat | Location | Source |
|---|---|---|---|---|
| 225003 | 9.83 | 54.84 | Flensburg Firth, Gelting Bay | LLUR-SH |
| DMU547 | 10.09 | 55.67 | Little Belt to Northern Belt Sea | HELCOM / ICES |
| TF12 | 11.57 | 54.31 | Bay of Mecklenburg | IOW |
| DB2 | 12.88 | 54.40 | Darsser Bodden, "Sundische Wiese" | LUNG-MV |
| TF113 | 13.5 | 54.93 | Arkona Sea | IOW |

The land use to the south and west is dominated by agriculutral activities, which lead to nutrient inputs into the Baltic Sea via rivers and the atmosphere. The population density is lower than along the southern North Sea but still high inducing the input of various types of pollutants – i.e. organic pollutants, heavy metals, and plastic litter. The shipping traffic volume is high because a major European shipping route leads through this region connecting habors in the Baltic Sea, e.g. St. Petersburg, to the North Sea and more distant locations. Hence, the deposition of atmospheric shipping emissions and direct discharges of ships import pollutants and nutrients into the Baltic Sea.

## 2.4 Model Validation

The biogeochemical model results were validated against measurements. Surface water concentrations of dissolved inorganic nitrogen (DIN), dissolved inorganic phosphorus (DIP), and chlorophyll-a were considered for this purpose. Modeled concentrations of particulate organic nitrogen (PON) are also considered but not compared to measurements. Although it is denoted as PON, only the bioavailable PON is considered. Measurement data for model validation were taken from four sources:

- Measurement database of the Leibniz Institute for Baltic Sea Research (IOWDB)

- State Agency for Agriculture, Environment and Rural Areas Schleswig-Holstein (LLUR-SH)

- State Agency for the Environment, Nature Conservation and Geology of Mecklenburg-Western Pomerania (LUNG-MV)

- HELCOM oceanographic measurement database hosted by the International Council for the Exploration of the Sea (ICES)

A statistical validation of the model results with measurement data is difficult because the number of observations is limited – far below one measurement per month at most stations. Therefore, a visual validation of time series plots was done. Five stations were chosen to be presented in the results section. They are listed in Table 2 and marked in the map of Fig. 4. Geographical features mentioned in the station descriptions below are provided in Fig. 3.



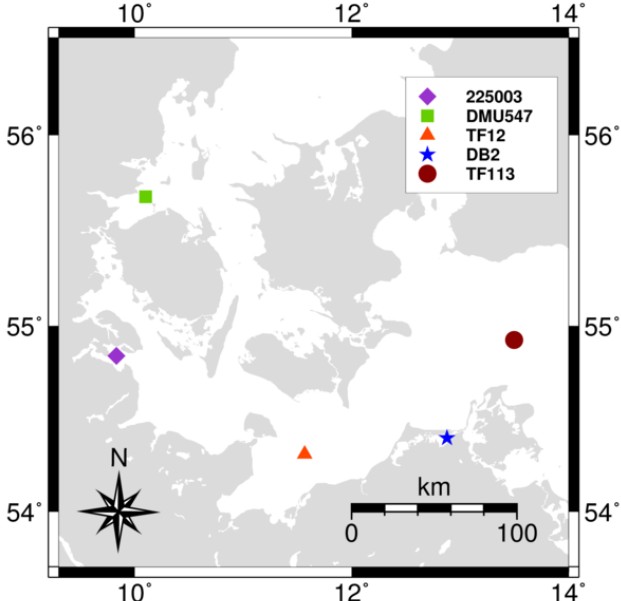

**Figure 4.** Locations of stations, at which measurement and model data are compared.

Station 225003 is located in the west of the study region and is surrounded by land (less than 10 km) towards North, West, and South. Due to its vicinity to land, we expect a considerable impact by land-sea-artifacts in the nitrogen deposition fields.

Station DMU 547 is located between the island of Funen and the Danish mainland at the boundary between Little Belt and Northern Belt Sea. Because westerly winds are dominating, this station receives atmospheric nitrogen, which predominantly
originates from sources on the Danish mainland, such as emissions from agricultural and transportation sectors.

Station 225003 is located in the east of the study region and is surrounded by land (less than $10\,\mathrm{km}$) by approximately $3/4$ towards North, West, and South. Due to its vicinity to land, we expect a considerable impact by land-sea-artifacts in the nitrogen deposition fields.

Station DMU 547 is located between the island of Funen and the Danish mainland at the boundary between Little Belt and
Northern Belt Sea. Because westerly winds are dominating, this station receives atmospheric nitrogen, which predominantly originates from sources on the Danish mainland, such as emissions from agricultural and transportation sectors.

Station TF12 is located in the center of the Bay of Mecklenburg. It should represent the open sea biogeochemistry in this bay well and should not be as strongly impacted by land-sea artifacts in the nitrogen deposition as the stations 225003, DMU547, and DB2.

Station DB2 is located in the Grabow, which is a lagoon-like water body and part of the Darss-Zingster Bodden Chain. Small rivers, which are not represented in the HBM-ERGOM model setup, drain into this lagoon. Through a small passage in the east, it is connected to the Baltic Sea. We expect a low performance of the model at this station because of its enclosed location. This is discussed in the Results and Discussion section.



Station TF113 is located in the center of the Arkona Sea. This station should be least impacted by land-sea-artifacts in the nitrogen deposition. It is closely located to the shipping route connecting the North Sea and the eastern Baltic Sea.

## 3 Results and Discussion

The benefit of high resolution nitrogen deposition data for biogeochemical model simulations is evaluated. We start with a brief
description of the used nitrogen deposition data sets (Sect. 3.1). A detailed discussion of these data sets is given in Karl et al. (in prep. same spec. issue a). In Sect. 3.2, we evaluate the impact of the different deposition data sets on the biogeochemistry and identify regions of particular importance. Section 3.3 provides an analysis of the shipping contribution.

### 3.1 Nitrogen Deposition

Figure 5 shows the annual mean nitrogen deposition in 2012 from the three deposition data sets. Oxidized and reduced nitrogen
are not distinguished. Generally, the deposition is highest along the coastline and decreases with increasing distance to the coast. There are three reasons for this: first, major nitrogen emission sources are located on the land. Nitrogen compounds with short atmospheric residence times are deposited in the vicinity of their sources. Second, gaseous nitrogen compounds, i.e. nitric acid and ammonia, condense on coarse sea salt particles, which are mainly emitted in the surf zone, have a high dry deposition velocity, and lead to a fast removal of condensed compounds from the atmosphere. Finally, higher nitrogen deposition along
the coast line can be an artifact: the dry deposition velocity of nitrogen species is higher above land than above water. The dry deposition is individually calculated per compartment at the ground. But, the model output of coastal model grid cells, which contain land and water surface, contains summed deposition onto both compartments. For the ecosystem model input processing, this deposition is split into marine and land deposition based on the known land-sea-fraction for each coastal grid cell. This leads to an overestimation of the marine nitrogen deposition and an underestimation of the nitrogen deposition onto
land. One could do the splitting based on the dry deposition velocities. These are, however, not constant in space and time but depend on other parameters such as air temperature. Hence, one needs to invest some work to do it properly, which is commonly – and also in this study – not done.

Comparing cmaq16 and cmaq04 deposition shows that the fine model resolution yields a higher degree of patchiness. This is caused by the finer resolved meteorological input data – particularly, by the convective precipitation (Karl et al., in prep. same
spec. issue a). This partly leads to higher nitrogen wet deposition in the cmaq04 case in some regions. This happens southward of Funen, where a precipitation front yields considerable nitrogen wet deposition on the fine resolved domain on 12th and 13th June. This region is clearly recognized as a large area in the nitrogen deposition difference plot (Fig. 8, further below). In contrast, the deposition along the coastline is lower in the cmaq04 case in most coastal regions. At a few coastal locations, the fine resolution yields higher nitrogen deposition. However, these locations, such as the Grabow, are harder to identify on the
map shown and are negligible by number and size.

One reason for these differences are changes in the representation of the coastlines in the two simulations. The water surfaces of the Ise and Roskilde Fjords were not resolved in the cmaq16 case leading to overestimated nitrogen deposition into these





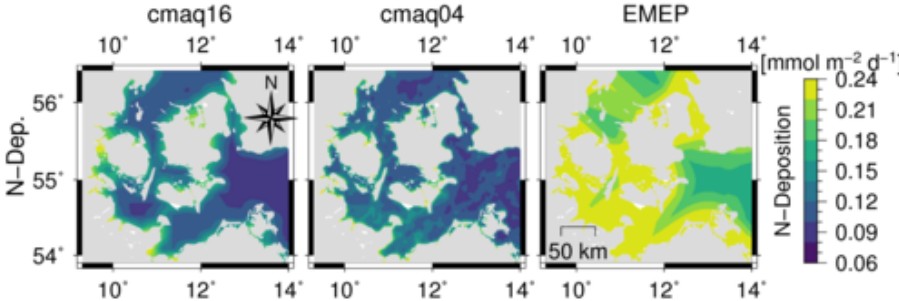

**Figure 5.** April to September mean nitrogen deposition of 2012 from three different nitrogen deposition data sets interpolated onto the HBM-ERGOM model grid resolution via bilinear interpolation. From left to right these are CMAQ coarse ($16 \times 16$ km$^2$), CMAQ fine ($4 \times 4$ km$^2$), and EMEP ($50 \times 50$ km$^2$).

water bodies in the cmaq16 case. In contrast, the land surface of the peninsula of Darss-Zingst was not resolved in the cmaq16 case leading to a lower nitrogen deposition in this case.

Model cases with higher nitrogen deposition might intensify phytoplankton growth in regions, where nitrogen is the limiting nutrient. The higher degree of patchiness in the deposition probably vanishes in the concentrations of the marine biogeochemical tracers due to natural variability and ocean currents.

The nitrogen deposition of the EMEP data set is considerably higher. Karl et al. (in prep. same spec. issue b) compares the nitrogen deposition of the two CMAQ simulations and a finer resolved EMEP model run ($0.1° \times 0.1°$ horizontal resolution) and has qualitatively similar findings. The spatial structure is less detailed, which is caused by the coarse spatial model resolution of $50 \times 50$ km$^2$. The plot does not show regular boxes but a less regular pattern because the data were interpolated in the target grid via bilinear interpolation. Whether this is the correct approach to interpolate from $50 \times 50$ km$^2$ to approximately $0.9 \times 0.9$ km$^2$ is debatable. But this discussion goes beyond the scope of this publication.

Since the EMEP nitrogen deposition is much higher, we might expect considerably higher biomass production. Particularly in summer, when nutrients are depleted in the marine water, the higher atmospheric input might yield a different system behavior.

## 3.2 Impact of the refined model grid resolution

### 3.2.1 Station time series, validation

Figure 6 shows the DIN, DIP, and chlorophyll-a concentrations of the three model cases at five locations. At most stations, measurement data for these parameters were available.

The DIN measurements are well met by the model cases in summer when nutrients are nearly depleted. In late winter and spring, the DIN concentrations are underestimated and the EMEP case yields the best results. Possibly too few nitrogen is provided in the model runs. However, it is not clear whether the nutrient recycling or the nitrogen inputs are underestimated.



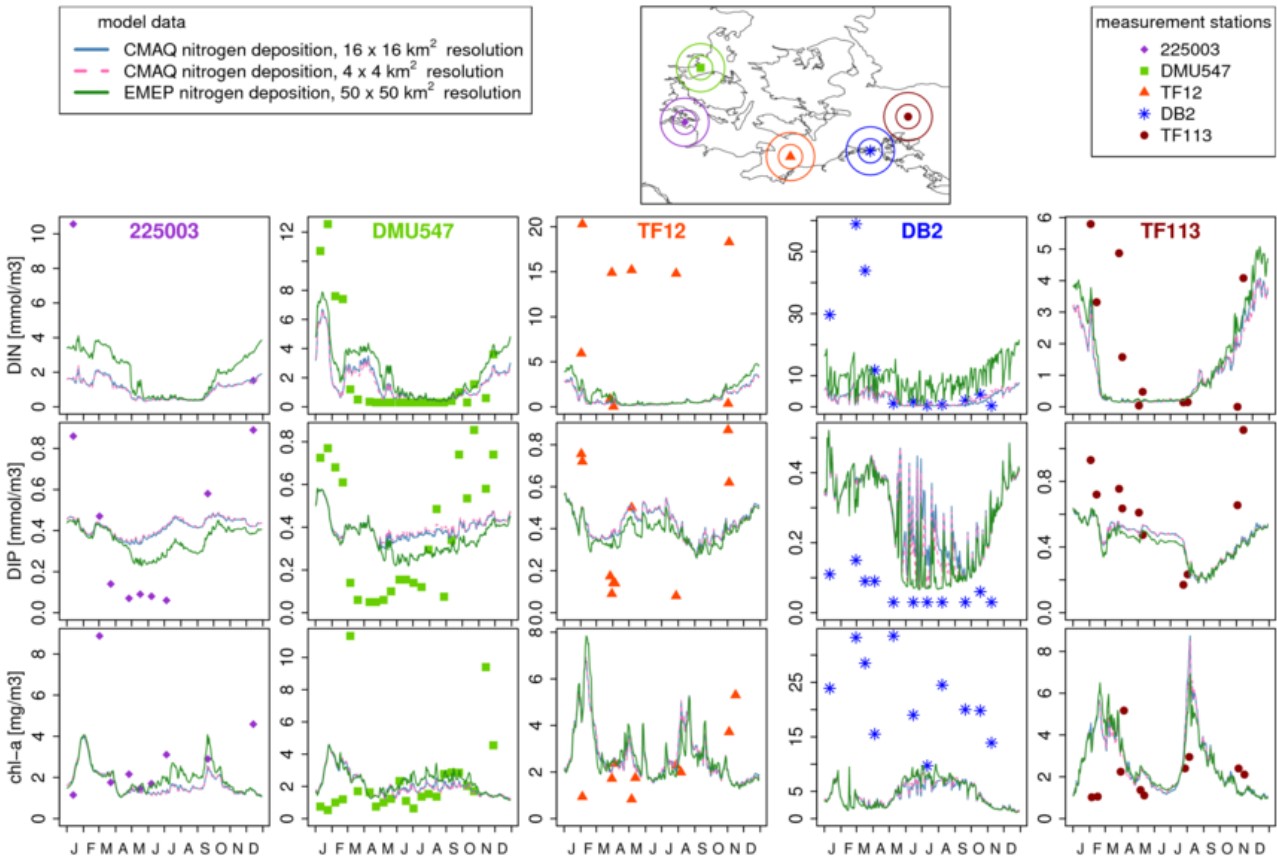

**Figure 6.** Comparison between modeled and measured surface layer concentrations at five stations. Measurement data are indicated by colored symbols (legend to right) and model data by colored lines (legend top left).

At dmu547, the model runs indicate peak DIN concentrations in spring, which are not measured. Additionally, at this station, the modeled increase of DIN in autumn seems to be too steep.

The DIP concentrations are overestimated at the stations 225003, dmu547, and tf12 in summer. Thus, either too few DIP is consumed by phytoplankton or too much DIP is in the model system. In winter, the DIP concentrations are underestimated

5  at stations 225003, dmu547, tf12, and tf113. Thus, the consumption and release of DIP might not be reproduced properly. Possibly, higher winter DIN concentrations would yield DIP depletion in summer as measurements at the stations 225003, dmu547 and tf12 indicate. At DB2, the DIP concentrations are overestimated throughout the year and oscillate considerably in summer. The oscillations are caused by occasional inflow events from the Baltic Sea into the Grabow, which transport DIP into the lagoon (not shown). Due to the low temporal resolution of the measurements it is not clear whether similar oscillations

10  arise in the real world.

The annual average chlorophyll-a concentrations are well reproduced at all stations except at db2. However, the temporal occurrence of chlorophyll-a peaks is not well represented. An algal bloom is predicted at the stations 225003, dmu547, and tf12



in January and February but it is not visible in the measurements. Vice versa, high chlorophyll-a measurements in November and December are not reproduced by the model at these stations. The measured chlorophyll-a concentrations are very high at db2 and by far underestimated in the model runs. Hence, too few biomass is produced in the Grabow leading to excess supply of DIP and to low chlorophyll-a concentrations.

The station DB2 is a special case as already noted in the Materials and Methods section. Because this study's ERGOM is set up to model biogeochemical processes in the Baltic Sea and in the North Sea, silicate is included as a nutrient. Commonly, silicate is present in excess in the Baltic Sea due to sufficiently large silicate input and recycling. However, the input of silicate into the Darss-Zingst Bodden Chain is too low because neither rivers flow into it (in the model) nor is the silicate inflow from the Baltic Sea sufficiently high. Therefore, silicate is depleted in the water column and in the sediment and no diatom bloom

evolves in spring explaining the lack of high chlorophyll-a concentrations in this season.

### 3.2.2    Station time series, deposition case comparison

The cmaq04 and cmaq16 cases show minor differences compared to each other, while the emep case underlines that changes in the atmospheric deposition impact the biogeochemical cycles. The DIN concentrations of the cmaq16 case are slightly higher at the stations 225003, dmu547, and tf12. They are lower at DB2 and about equal at tf113. In contrast, the DIP concentrations

are slightly lower at 225003, dmu547, and tf12 during summer. This indicates a higher primary production in summer caused by the higher nitrogen supply. At DB2, the DIP concentrations oscillate strongly in summer and peak concentrations differ. The oscillations were discussed above. The chlorophyll-a concentrations are very similar between the cmaq04 and cmaq16 cases.

Compared to the EMEP case, the DIN concentrations are very similar in both cmaq model cases during summer. During the first three and the last three months of the year, the DIN concentrations are considerably higher in the EMEP case. DIP

concentrations, in contrast, are much lower in the EMEP case especially during summer. Accompanied by slightly higher chlorophyll concentrations, this indicates a higher primary production in summer due to higher DIN concentrations in winter. Nevertheless, the difference in chlorophyll-a is much lower: at the station 225003, the summer chlorophyll-a concentrations are higher in the EMEP case, whereas at the other stations our three cases are close to each other.

The vicinity to land correlates with the highest differences between the three model cases: the station 225003, which is

closely surrounded by land, reveals the highest differences. The DIN and DIP concentrations of the three cases get closer to each other with increasing distance to the coast. And, the station tf113 shows the least differences in DIN, DIP, and chlorophyll-a concentrations among the considered stations.

### 3.2.3    Spatial evaluation

Figure 7 shows the mean dissolved inorganic nitrogen (DIN, top row) and particulate organic nitrogen (PON, bottom row)

concentrations of the cmaq16, cmaq04, and EMEP cases (left to right) in the surface layer averaged over April to September 2012. Although it is denoted as PON, only the bioavailable PON is considered.

In the cmaq04 and cmaq16 cases, the highest DIN concentrations arise eastward of the Danish mainland, in the Ise and Roskilde Fjords, in the Darss-Zingst Bodden Chain (west of Rügen), and in the Greifswald Bay. The high DIN concentrations



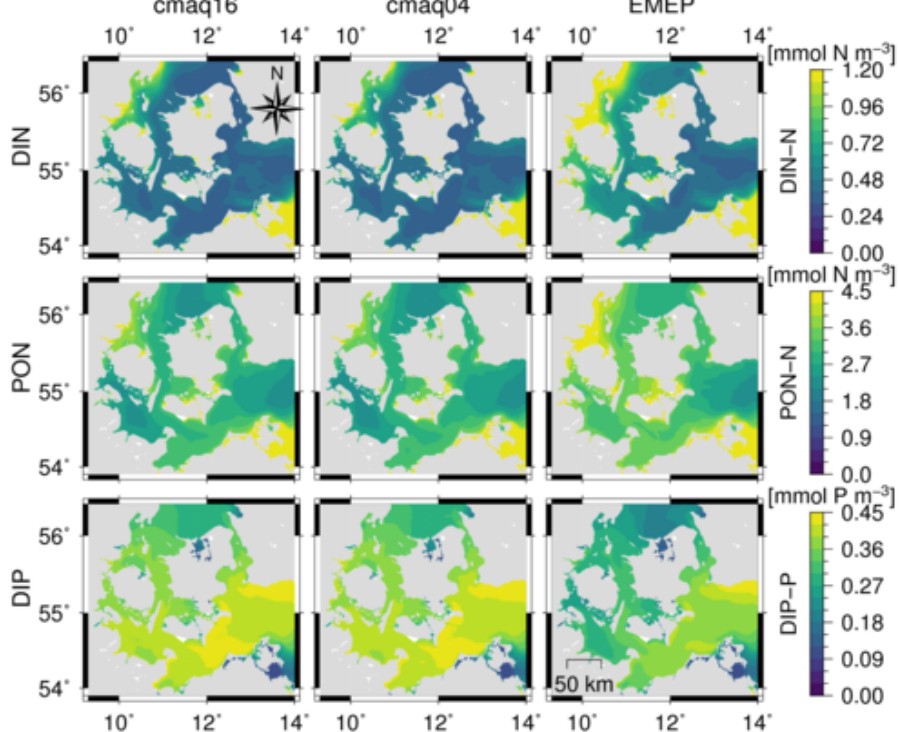

**Figure 7.** April to September mean concentrations dissolved inorganic nitrogen (DIN, top row), particulate organic nitrogen (PON, center row), and dissolved organic phosphorus (DIP, bottom row) concentrations of the cmaq16, cmaq04, and EMEP model cases in the surface layer. PON in this context means the bioavailable PON.

eastward of the island of Rügen originate from the Oder river. The DIN concentrations in the Bodden Chain do not originate from rivers (no rivers are included in these region) but are due to atmospheric deposition of nitrogen (see Fig. S.1, Supplement). So are the DIN concentrations in the Ise and Roskilde Fjords. Notably, the spatial pattern of the DIN concentrations does not fully correlate with the pattern of the nitrogen deposition (Fig. 5). The PON concentrations are smoother than the DIN
5 concentrations. They also take maximum values in the region around Rügen and in the Darss-Zingst Bodden Chain. Maxima in other coastal regions are less pronounced.

In the EMEP case, the DIN concentrations are much higher, particularly along the Danish coast. The spatial pattern is similar to the pattern of the other cases – except for the magnitude. The PON concentrations are only slightly higher than in the other cases. This is reasonable because phytoplankton growth does not only depend on nitrogen but it is also limited by
10 the availability of other nutrients and by environmental conditions, e.g. temperature and vertical stratification. This becomes obvious as DIP concentrations are considerably reduced in the EMEP case compared to cmaq16 as the DIP differences in Fig. 8 (bottom row, columns 3 and 4) show.



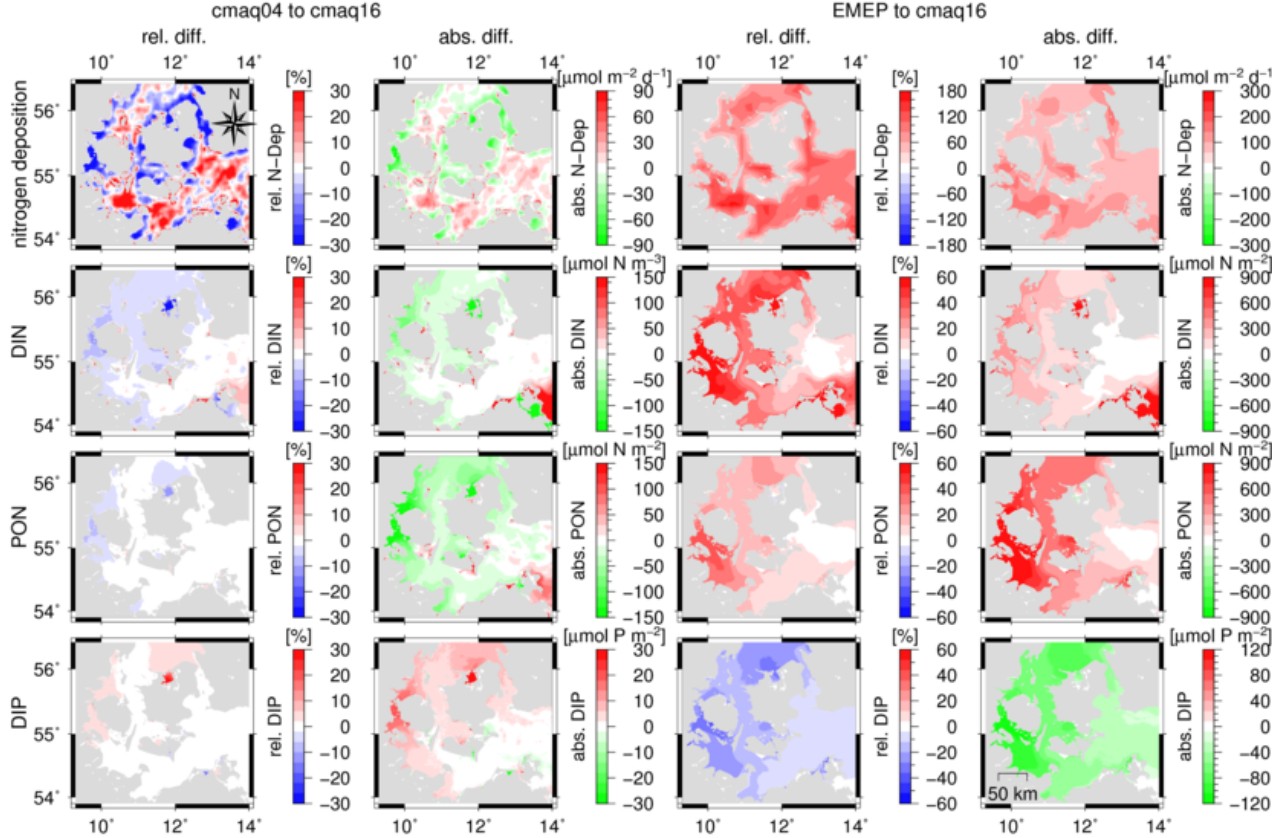

**Figure 8.** Differences between cmaq04 and cmaq16 on the left (columns 1 and 2) and between EMEP and cmaq16 on the right (column 3 and 4). Columns 1 and 3 show relative differences with respect to cmaq16 and columns 2 and 4 show absolute differences. These values are means of nitrogen deposition, DIN, PON, and DIP (top to bottom) over a period from April to September. PON in this context means the bioavailable PON.

The differences in the spatial distributions between cmaq04 and cmaq16 cases are low and difficult to evaluate on the basis of Fig. 7. Figure 8 shows the relative and absolute differences of the cmaq04 case with respect to the cmaq16 case in two columns on the left. Notably, the pattern shown of absolute differences is rather patchy. The DIN concentrations rise in the Darss-Zingst Bodden Chain but they decrease in the Ise and Roskilde Fjords. This result was to be expected from the comparison of the deposition data. The DIN concentrations in the Little Belt, which is also impacted intensely by atmospheric nitrogen deposition, decrease as well. The DIP concentrations increase considerably due to lower DIN concentrations in the Ise Fjord and in the Little Belt. In the open water, no clear pattern is visible although the differences in the nitrogen deposition are rather patchy.

For PON, the relative difference between cmaq04 and cmaq16 is lower than for DIN and the spatial distribution is smoother. Particularly along the German coast, the differences are smaller. In a few coastal regions, especially in the Darss-Zingst Bodden



Chain, a silicate limitation for diatom growth arises as described in Sect. 3.2.1: as a result, the PON concentrations are considerably damped. In contrast, the PON concentrations in the Roskilde Fjord are changed by the same amount as the DIN concentrations. Here, no silicate limitation arises. In the open waters, a spatial pattern becomes visible in the relative differences. It does not correlate with geographic features but rather arises from differences in precipitation fronts of the meteorological

forcing or from biogeochemical processes.

Summarizing the spatial variation, the impact of a refined model resolution on the biomass production is rather low in most regions of the domain. Some coastal areas showed high relative differences in the DIN and PON concentrations of the cmaq04 and cmaq16 cases. However, in some of these areas, the quality of the model output is questionable because of missing river inflows or other model-related problems such as silicate limitation. Additionally, the variation caused by refining the model

resolution (cmaq16 to cmaq04) is much lower than the difference to another nitrogen deposition data set – namely EMEP.

None of the model cases revealed a different behavior of the biogeochemical system. Thus, the system dynamics seem to be quite stable with respect to changes in the model input. Atmospheric deposition provides approximately a quarter to a third of the nitrogen input into the Baltic Sea (HELCOM, 2013a, b). However, the residence time of nitrogen and phosphorus in the water column of the Baltic Sea is in the order of several years (Radtke et al., 2012). The residence time of phosphorus exceeds

the residence time of nitrogen. A longer simulation on the order of ten years or more would be required to include possible long-term effects of the differences in nitrogen deposition into our analysis.

### 3.3    Shipping Contribution

Figure 9 shows the DIN and PON concentrations resulting from ship-related nitrogen deposition. The left column contains results of the cmaq16 case and the center column results of the cmaq04 case. In the right column, the ratio of shipping nitrogen

to total nitrogen is shown for the cmaq16 case. A corresponding plot for the cmaq04 case looks quite similar and is not shown.

Hotspots of shipping-related DIN concentrations are in the Ise and Roskilde Fjords, eastward of Rügen, and in the Darss-Zingst Bodden Chain. A few small bights along the Danish islands also show high shipping DIN. However, shipping-related DIN remains below $0.1 \, \mathrm{mmol \, m^{-3}}$ in most regions. Generally, the ratio of shipping DIN to total DIN (Fig9 top right) is not dominant but exceeds $5\,\%$ in more than $50\,\%$ of the surface waters. In peak regions even $10\,\%$ is exceeded. The shipping-to-

total-DIN ratio further shows indirectly that the shipping-related nitrogen deposition spatially differs from the total nitrogen deposition. This is reasonable because the shipping emissions primarily take place along the major shipping routes. Areas with a high shipping contribution are the Ise and Roskilde Fjords, southward of the island of Funen, at the southern tip of Zealand, and in the Darss-Zingst Bodden Chain.

The shipping PON is similarly distributed as the shipping DIN. The ratio of shipping PON to total PON is lower than the same

ratio for DIN. This clearly shows, that the short-term impact of atmospheric nitrogen deposition on phyto- and zooplankton is lower than one would expect without applying a biogeochemical model.

The spatial pattern of the DIN and PON concentrations shows only minor differences between the cmaq16 and cmaq04 cases. To better evaluate these differences, Fig. 10 shows the relative difference between the cmaq04 and cmaq16 cases with respect to shipping related nitrogen. The spatial pattern reveals a higher degree of patchiness than the same figure for total nitrogen



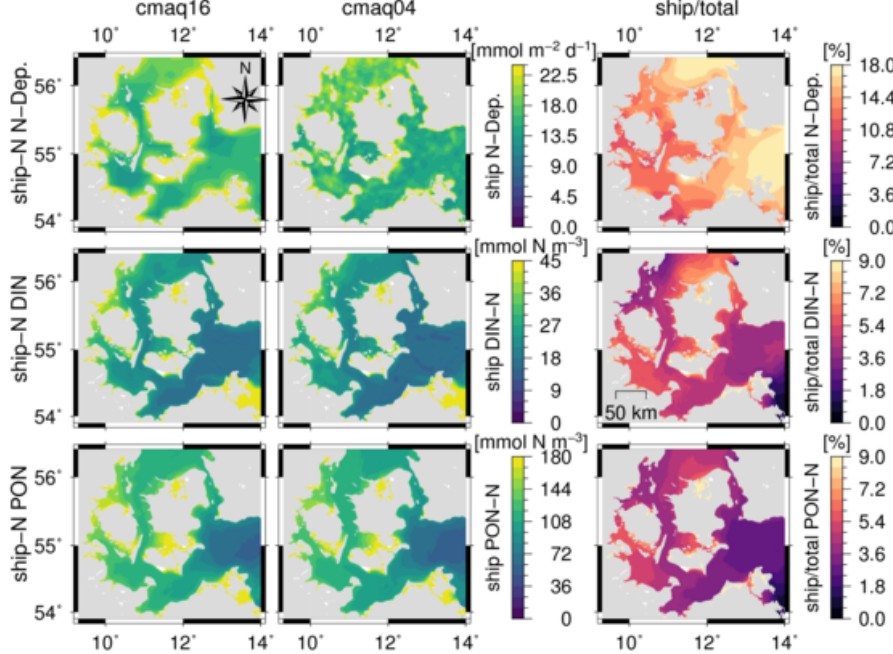

**Figure 9. left and center column:** similar to Fig. 7 but only for shipping related nitrogen; nitrogen deposition (top row), DIN (center row) and PON (bottom row) concentrations in the surface layer from shipping-related nitrogen of the cmaq04 (left) and cmaq16 (center) cases. **right column:** ratio of the shipping-related nitrogen to total nitrogen in nitrogen deposition, DIN and PON of the cmaq16 case. The same plot for the cmaq04 case is not shown but quite similar. PON in this context means the bioavailable PON.

deposition (Fig. 8). In the cmaq04 case, the shipping contribution is considerably increased in a large region southward of Funen (large red patch), in the Darss-Zingst Bodden Chain, and at a few other coastal locations. It is reduced everywhere else and particularly in the northern part of the Little Belt, north of Lolland, in the Ise Fjord, around Rügen, and close to the Trave. A

5   relatively high shipping-nitrogen contribution in combination with the vicinity to land is common to all these regions. The large red patch southward of Funen originates from a precipitation front, which was resolved by the meteorological model on the fine grid but not on the coarse one (see Sect. 3.1). The red region in the Bodden Chain arises due to averaging interval from April to September as a view on the time series at the stations db2 indicates (Fig. S.2, Supplement): shipping DIN concentrations are higher in the cmaq16 case from January to April and from November to December but higher in the cmaq04 from Mai to October.

10      Summarizing, the shipping sector is a relevant contributor for surface DIN and PON concentrations. Increased resolution of atmospheric models has a stronger impact on the shipping related nitrogen contribution than on the total nitrogen. However, the total nitrogen and not only the atmospherically contributed nitrogen was considered in Sect. 3.2, whereas in Sect. 3.3 the shipping-related atmospheric input was evaluated. Evaluating the total atmospheric contribution to DIN might have probably





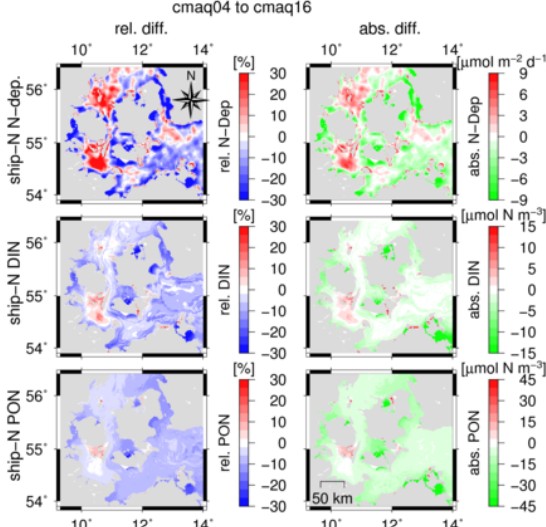

**Figure 10.** Similar to Fig. 8 but only for shipping-related nitrogen. Relative (left) and absolute (right) differences for shipping-related nitrogen between cmaq04 and cmaq16 for April to September mean of the nitrogen deposition (top) and DIN (center) and PON (bottom) concentrations in the surface layer. Relative differences are calculated with respect to cmaq16. PON in this context means the bioavailable PON.

provided a qualitatively similar picture than this shipping evaluation. Notably, the spatio-temporal pattern of the biogeochemical tracers does not change much, when higher resolution input data is employed.

## 4 Conclusions

This study had two major aims:

5     1. What resolution of atmospheric nitrogen deposition is necessary to properly evaluate the impact of nitrogen deposition on DIN concentrations and biomass production in highly structured coastal areas?

    2. How high is the contribution of shipping related nitrogen deposition to this?

    HBM-ERGOM model simulations with three nitrogen deposition data sets were performed for the first assessment. Two datasets were produced with the same chemistry transport model (CTM), i.e. CMAQ, but with different model resolutions and 10  driven by two differently resolved meteorological data sets. The coarse resolution data set ($16 \times 16$ km$^2$, "cmaq16") yielded higher nitrogen deposition in most coastal areas than the fine resolved one ($4 \times 4$ km$^2$, "cmaq04"). The fine CMAQ model run was forced by a finer resolved CCLM model run, in which smaller scale precipitation events were resolved. That latter fact led to a higher degree of spatial patchiness of the nitrogen deposition above open waters and partly to higher nitrogen wet deposition in the cmaq04 case. A third data set, EMEP nitrogen deposition data of the 2016 reporting period was chosen



(50 $\times$ 50 km$^2$ resolution) to get a feeling for the inter-model variability. Independent of the coarser resolution, the EMEP deposition was approximately twice as high as the CMAQ nitrogen deposition in the study region. Hongisto (2014) compared their nitrogen deposition calculations with EMEP data for the whole Baltic Sea. Their reduced nitrogen deposition was 5 % to 22 % lower than that of the EMEP model. The oxidized nitrogen was of similar magnitude. Hongisto (2014) accounted for

the differences to the different resolution of the coastline. In the region considered in this study, the coastline has an even higher impact than in Hongisto (2014) because less open ocean areas are covered and most 50 $\times$ 50 km$^2$ EMEP grid cells are expected to contain water and land. The evaluation of the nitrogen deposition data was not part of this study. Karl et al. (in prep. same spec. issue a) compared the two CMAQ model runs and a finer resolved EMEP model run with each other. A validation of the nitrate deposition with measurements of the "Cooperative Programme for Monitoring and Evaluation of the

Long-range Transmission of Air Pollutants in Europe" (EMEP) monitoring network can be found in Karl et al. (in prep. same spec. issue a). Due to a lack of deposition measurements over the Baltic Sea, no validation of the nitrogen deposition data with measurements over marine regions was performed. Due to a lack of deposition measurements over the Baltic Sea, no validation of the nitrogen deposition data with measurements over marine regions was performed.

The model output was compared with observations of DIN, DIP, and chlorophyll-a at several stations in the model domain.

Data for five stations were presented in this publication. All three model cases developed a phytoplankton bloom in spring, depletion of DIN, and a strong decline of DIP. By the end of the year, the surface DIN and DIP concentrations rose again. Chlorophyll-a concentrations were in an acceptable order of magnitude but lacked the correct temporal occurrence of peak concentrations.

Improving the resolution of the atmospheric model from 16 $\times$ 16 km$^2$ to 4 $\times$ 4 km$^2$ did not yield obvious improvements

with respect to the measurements. Hence, the spatial characteristics of the atmospheric deposition had only a minor impact on the reproducibility of measurements by the model – partly due to the scarcity of measurement data. The EMEP case led to considerably higher DIN concentrations in winter and lower DIP concentrations in summer. Nevertheless, none of the model cases agreed with the measurements to a significantly higher degree than the others. The only exceptions were DIN concentrations in spring, which are best realized by the EMEP case. Differences between DIN concentrations by the

three model cases were most pronounced in winter, whereas differences between DIP concentrations arose in summer. These differences might increase over years. Also, none of the model cases revealed a different behavior of the biogeochemical system. Because only one year of model simulations was evaluated, it might be reasonable to look into longer time periods in future studies. In summary, none of the deposition data sets can be graded as "better suited" for short-term ecosystem model simulations.

In addition to the time series evaluation, the spatial distribution of April-to-September-mean concentrations of DIN and PON were considered. The spatial pattern of the DIN and PON concentrations was qualitatively similar in all three model cases. The PON concentrations were spatially smoother than the DIN concentrations. Quantitatively, the EMEP case yielded the highest DIN concentrations. The PON concentrations of the EMEP case were also higher but relatively close to the other cases' PON concentrations than DIN. The cmaq16 and cmaq04 cases were quite similar to each other and revealed major differences only

in coastal regions. Mainly, the coarse resolution led to higher DIN concentrations – with only a few exceptions, such as the





Darss-Zingst Bodden Chain. Unfortunately, the biogeochemistry in the area of this lagoon was not well reproduced by the model, wherefore, the anomaly was not evaluated in detail. At the open ocean, minor patchy differences between the cmaq16 and cmaq04 cases were identified. These were induced by differently resolved precipitation events in the driving meteorology leading to differences in the nitrogen wet deposition. However, the relative differences in the DIN and PON concentrations

were considerably lower than the relative differences in the nitrogen deposition.

To summarize, refining the CTM resolution from $16 \times 16$ km$^2$ to $4 \times 4$ km$^2$ brought no apparent benefit in most regions. However, close to the coast and in Fjords, Boddens, and Lagoons it seems to be important. Unfortunately, the quality of the HBM-ERGOM model results was very low in these coastal regions because small inflowing rivers were missing in the model setup and marine coastal processes are generally more difficult to reproduce than open-ocean biogeochemistry. Particularly

the interaction of the water column with the sediment and nitrification-denitrification processes at the water-sediment interface were probably not fully covered (Asmala et al., 2017). Thus, the benefit of improving the quality of nitrogen deposition could not be verified in the ecosystem model results.

Additionally, it has to be noted that the difference between different atmospheric deposition products (EMEP vs. cmaq04/16) is much higher than the difference between the coarse and fine resolution of one product (cmaq04 vs. cmaq16). Due to the large

model-related differences between CMAQ and EMEP nitrogen deposition, it was not even possible to determine whether going from $50 \times 50$ km$^2$ to $16 \times 16$ km$^2$ resolution yields an improvement. Therefore, emphasis should be put on the improvement of the nitrate formation scheme and the nitrogen deposition parameterizations in the atmospheric models themselves rather than refining model resolution. Alternatively, using ensemble model nitrogen depositions as a driver for marine biogeochemical models might be expedient. To evaluate EMEP and cmaq16 nitrogen deposition, one should rather focus on marine regions

with less coastline in future studies such as the eastern Gotland basin.

Running very high resolution meteorological and chemistry transport simulations is considerably more computing resource intensive than doing the same for a sixteen times coarser resolution. Hence, reduced model resolution will consume less computing time and result in a lower power consumption of HPC clusters.

One could observe that the choice of the atmospheric deposition data set had a surprisingly low impact on the ecosystem

dynamics and concentrations of biogeochemical tracers. A reason for this might be the long residence time – in the order of several years – of nitrogen and phosphorus in the water column of the Baltic Sea (Radtke et al., 2012). Therefore, the atmospheric deposition of one year has only a low contribution to the nitrogen budget of the Baltic Sea. In other regional seas with shorter residence times of nutrients, such as the North Sea, one might expect a stronger impact of different atmospheric deposition data sets. Hence, one should not generalize from these study's results on other marine regions and, instead, evaluate

this aspect in future studies. Moreover, simulation of longer time scales should be performed in future studies until the tagged atmospheric nitrogen reaches a steady state in the region of interest. Only a short time span of two years (one year spin-up + one year actual model run) was simulated in this study because HBM-ERGOM is not made for highly parallelized multi-decadal simulations: the model is made for operational forecasts on small computing clusters and, hence, is not optimized for highly parallelized simulations.





The shipping sector is an important contributor to atmospheric nitrogen deposition into the Baltic Sea. Previous studies found that up to $15\,\%$ of the oxidized nitrogen deposition (Tsyro and Berge, 1998; Bartnicki and Fagerli, 2008; Bartnicki et al., 2011) and around $6\,\%$ of the total nitrogen deposition (HELCOM, 2005) into the Baltic Sea were caused by the shipping sector. These numbers have to be put in context because only $1/4$ to $1/3$ of the nutrient input into the Baltic Sea originates from the

atmosphere. Nonetheless, these numbers are not negligible.

In this study, the focus was not on the atmospheric input of shipping nitrogen but on the contribution of this nitrogen to the biogeochemical processes in the time frame of one year. Such an evaluation is important, when benefits of measures for reductions of the nutrient input are assessed. Exemplary, one might want to assess the benefit of Nitrogen Emission Control Areas (NECAs) on the water quality as $NO_X$ emissions per ship are expected to decline through this legislation in the next

decades.

In summary, the shipping sector is a relevant contributor for surface DIN and PON concentrations with each up to $10\,\%$ in peak regions. Some coastal areas are particularly impacted. This might be due to high shipping traffic in these regions. Refining the resolution of CTM had a stronger impact on the shipping related DIN and PON than on the total DIN and PON. Based on these findings, $NO_X$ emission reduction measures, i.e. NECAs, might reduce the nutrient load in hot-spot regions by a few

percent. Karl et al. (in prep. same spec. issue a) presents and discusses results from NECA scenario in 2040.

To assess whether reducing ship borne nitrogen emissions is an appropriate approach to counteract marine eutrophication, one needs to know how much nutrients are contributed by other emission sectors to the marine oxidized and reduced nitrogen deposition. Therefore, in future studies other emission sectors such as road transport, industrial combustion, and agricultural activities should be evaluated. Moreover, simulations over longer times spans – i.e. ten to twenty years – should be performed

to cover the cumulative effect of small annual reductions. Simulations should be performed until a steady state is reached as noted further above.

## 5   Outlook

Considerable research has been done on nitrogen deposition. There exists broad knowledge on emissions of nitrogen compounds and on the spatio-temporally patterns of nitrogen deposition. However, research on emissions of phosphorous com-

pounds and on their deposition into water bodies has received only low scientific interest. Particularly, emissions of phosphate containing dust from agriculturally used areas is hardly ever considered in modern atmospheric chemistry transport model simulations. As a result, a spatio-temporally constant phosphate deposition is assumed in this study. Considerable research is necessary for improving the knowledge and understanding in this field. In a future study, one might try to derive phosphate deposition fields from available dust forecasts and from the deposition of related compounds and calibrate these via phosphate

deposition measurements.

In this study, only one year was considered. However, the impact of different deposition data sets might take several years to develop its full impact. Therefore, future studies should focus on a longer time period – i.e. five to ten years when nitrogen is considered.



In this study, the contribution of the shipping sector to marine DIN and PON concentrations was considered. The residence time of nutrients is shorter in the North Sea than in the Baltic Sea. Therefore, atmospheric deposition might have a different impact on North Sea biogeochemistry and algal blooms. In future studies, one might look into the contribution of other atmospheric emission sources and compare their impact in the North Sea and Baltic Sea region.

*Code and data availability.* **Code:** The original HBM-ERGOM code was provided by the Federal Maritime and Hydrographic Agency of Germany (BSH). The license agreement does not allow the authors to pass the code to third parties. The code can be requested from the BSH or the Danish Meteorological Institute (DMI). The modified ERGOM code and brief description of the model processes and constants are attached in the supplement.

**Model output data:** The data are available via the THREDDS server of the IOW.

– cmaq16: https://thredds-iow.io-warnemuende.de/thredds/projects/meramo/catalog_meramo_cmaq16_nosilrestart.html

– cmaq04: https://thredds-iow.io-warnemuende.de/thredds/projects/meramo/catalog_meramo_cmaq04_nosilrestart.html

– EMEP: https://thredds-iow.io-warnemuende.de/thredds/projects/meramo/catalog_meramo_emep_nosilrestart.html

**Model input data:**

– The meteorological input data for HBM-ERGOM were created by the German Weather Service (DWD) for the BSH and kindly
provided by the BSH for this study. We are not allowed to publish them because they are under the license of the DWD.

– The EMEP nitrogen deposition data are available via the THREDDS server of Norwegian Meteorological Institute: http://thredds.met. no/thredds/catalog/data/EMEP/2016_Reporting/catalog.html

– The CMAQ nitrogen deposition data are available on request from the co-authors of the HZG.

**Measurement data:**

– data of state agencies (LLUR-SH and LUNG-MV): please contact René Friedland

– HELCOM data are available via the ICES homepage: http://ocean.ices.dk/helcom/Helcom.aspx

– IOWDB data are available on request (https://www.io-warnemuende.de/iowdb.html). Please contact authors to get access to the database.

*Author contributions.* Daniel Neumann was responsible for overall structure and for writing the manuscript. He performed the HBM-
ERGOM model simulations and did major programming and plotting tasks. Hagen Radtke implemented the tagging method and a tool for model validation. He contributed to the Materials & Methods and Results & Discussion sections. René Friedland provided measurement data, participated in the evaluation of the model data, and contributed to the Results & Discussion sections. Matthias Karl performed CMAQ air quality model simulations and evaluated meteorological forcing data and nitrogen deposition data. He contributed to the Materials & Methods and Results & Discussion sections. He further helped developing the research questions. Volker Matthias contributed to the
state of knowledge, to the Introduction section. To the development of the research question. He further provided input data for the CMAQ



model simulations. Thomas Neumann supported developing the research question and contributed to Introduction, Materials & Methods and Conclusions sections.

*Competing interests.* The authors declare that they have no conflict of interest.

*Acknowledgements.* Parts of the research published in this publication were carried out in the research projects MeRamo (funded by BMVI, FKZ 50EW1601), MOSSCO-Synthese (funded by BMBF, FKZ 03F0740B), and SHEBA (Sustainable Shipping and Environment of the Baltic Sea region, EU BONUS Project, Call 2014-41). The BONUS SHEBA project was supported by BONUS (Art 185), funded jointly by the EU and national funding institutions. The HBM-ERGOM model simulations were performed at the cluster Gottfried of the North-German Supercomputing Alliance (HLRN, project ID mvk00054) within MeRamo. The meteorological and atmospheric chemistry transport model (CTM) simulations were performed for SHEBA at the German Climate Computing Center (DKRZ) within the Project "Regionale Atmo-sphärenmodellierung" (Project ID 302), which is funded by the Helmholtz Association. The emission for the CTM simulations were kindly provided by Johannes Bieser, Armin Aulinger, and Jukka-Pekka Jalkanen. The HBM is currently maintained by the Danish Meteorological Institute (DMI) and the Federal Maritime and Hydrographic Agency of Germany (BSH). The air quality model is developed and maintained by the U.S. Environmental Protection Agency (US EPA). We thank our colleagues conducting IOW's Baltic Monitoring and long-term data program, which intense quality checked measurements we used for the model validation. Some of the measurement data were kindly provided by German environmental state agencies – namely the Agency for Agriculture, Environment and Rural Areas Schleswig-Holstein (LLUR-SH) and the Agency for the Environment, Nature Conservation and Geology of Mecklenburg-Western Pomerania (LUNG-MV) – and by the HELCOM oceanographic measurements database hosted by ICES. Jan Eiof Jonson and colleagues from the Norwegian Meteoro-logical Institute provided details on the EMEP model setup. Martin Schmidt of the IOW supported us with respect to preparation and upload of model data to the IOW THREDDS server. Bronwyn Cahill provided valuable comments to the manuscript. We thank Uwe Schulzweida, Charlie Zender, Paul Wessel, the R Core Team, and the Unidata development team (and all involved developers/contributors) for maintaining the open source software packages Climate Data Operators (cdo), the NetCDF Operators (NCO), Generic Mapping Tools (GMT), the sta-tistical computing language R, and netCDF, respectively.cdo), the NetCDF Operators (NCO), Generic Mapping Tools (GMT), the statistical computing language R, and netCDF, respectively.



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
