# Peer review of "Importance of high resolution nitrogen deposition data for biogeochemical modeling in the western Baltic Sea and the contribution of the shipping sector"

_Ocean Science, 2018_

## Referee Comment (RC1) · F Große (Referee) · 12 Sep 2018

**Review on "Importance of high resolution nitrogen deposition data for biogeochemical modeling in the western Baltic Sea and the contribution of the shipping sector" by Neumann et al.**

This study presents an analysis of the impact of atmospheric nitrogen (N) deposition, and in particular the contribution from ship-borne emissions, to the biogeochemistry in the western Baltic Sea. This region is affected strongly by atmospheric N deposition from land sources as well as from ships in the North Sea and Baltic Sea.

The authors use three different datasets (high to low spatial resolution) provided by two different atmospheric models (CMAQ and EMEP) to force their physical-biogeochemical model (HBM-ERGOM) to investigate the impact of different resolution and different data sources on the marine environment. They furthermore apply an element tracing method to the atmospheric inputs from the shipping and non-shipping sectors to quantify the contribution of the former to biogeochemical processes in the study region. A 2-year simulation (incl. a 1-year spin-up) is conducted for each of the different deposition datasets.

The authors find that the effect of different data products (i.e. atmospheric model outputs) is much larger than the effect of high vs. low resolution of the atmospheric deposition data. In addition, the contribution to the shipping sector to dissolved inorganic N (DIN) exceeds 5% in large parts of the study regions, with maximum values up to 10%. The contribution to particulate organic N (PON) is found to be slightly lower. Hence, the shipping sector plays a small, nevertheless relevant role for the biogeochemistry in the western Baltic Sea.

The manuscript is generally well written and easy to read. However, I think in its current form the manuscript lacks a strong conclusion/scientific finding owed to the study setup. In addition, some of the analyses could be more explicitly address the research questions defined by the authors.

My strongest criticism – as also mentioned by the authors themselves – is that the current study setup, in particular the comparably short simulation period, is likely not sufficient to draw general conclusions on the impacts of both model resolution and the shipping sector on the biogeochemistry in the study region. Instead of referring to future studies, the authors should address at least one of the aspects they outline in their introduction to a sufficient level in order to provide a strong study and conclusion. Besides this, I have a few minor points, which I would like to see addressed before considering the manuscript for publication (outlined in the comments below).

Therefore, I recommend reconsidering the manuscript for publication after major revision.

1. **Major points/General comments**

**Study setup and findings**

From my perspective, the current study does not allow for a strong conclusion on any of the points raised by the authors (model resolution, different datasets, shipping sector), which leads to Conclusions/Outlook, which rather read as a discussion of the limitations of the setup and a referral to future studies to overcome these. As discussed by the authors, this is due to the simulation period of only two years relative to the long residence times of N (several years) in the Baltic Sea. The former is owed to the fact that the applied marine model is not designed for multi-year simulations, as it is not sufficiently parallelized. Consequently, the model is not run to steady state.

I understand this technical constraint. However, there needs to be (at least) one strong and important finding that is sufficiently supported by the present study. This could be either the importance of resolution vs. different datasets or the impact of the shipping sector. The former might be easier to do, as there is already a strong indication that the different datasets overrule the effect of the different resolutions. However, it would be worthwhile to conduct this analysis based on a simulation that reached steady state, and thus provide not only qualitative indications but also a quantitative assessment of the impact. This would make the study much more relevant than it currently is.

Providing a comparable analysis (i.e. for a steady state) for the shipping sector's contribution would, of course, be very interesting as well, and even more relevant from a scientific point of view. However, I suppose it is computationally more expensive due to the element tagging, and the authors may want to choose only one of the two topics.

If neither of the two is feasible, the authors should state clearly in the introduction and conclusions that this study is only a proof of concept and more comprehensive analyses are required and left to future studies. However, this would significantly reduce the impact of the study. Hence, I recommend addressing one of the two topics as described above.

I further wonder why the authors chose the coarse (50km x 50km) resolution EMEP dataset although there is finer resolution (0.1° x 0.1°) EMEP data available? The resolution of the latter would be much closer to the CMAQ16 case, and thus would allow for a better analysis of the effects of resolution vs. different datasets. The results may not change significantly (Karl et al., in prep. same spec. issue b), nevertheless using this data also in the present study, would make it more consistent with the study goal.

**Influence on biogeochemical processes**

My second major point is that the study claims to analyze the impact of resolution, datasets and shipping sector on the "biomass production" (Introduction, page 3, first research question) or "biogeochemical processes" (Conclusions, page 22, lines 6-7). However, they do not show results of actual processes, but only DIN and PON (and DIP). Instead of presenting results for PON, an analysis of the effects/contributions to primary production (PP) would be much more insightful and would explicitly address the formulated research question. In this context, I would also suggest to not only analyze surface values (like in Fig. 9), but to consider water column averages (or water column integrals for PP). The differences between the shipping contributions to DIN and PON might simply be related to the sinking of (detritus as a part of) PON into deeper layers.

**2. Minor points/Specific comments**

Page 3, line 27: What is "this"? Biomass production? Please specify in the question.

Page 3, lines 28-34: It should be mentioned that a marine model is used in combination with different atmospheric deposition data (question 1) and element tracing (question 2).

Although, there is a well outlined motivation for this study, it gets a bit lost in the last paragraph of the introduction (page 4, lines 2-9), which reads like a rather long summary of existing research related to the present study. The authors could simply move the first sentence (page 4, lines 1-2) to the end of the paragraph to finish the introduction with a highlight statement.

Page 4, lines 11-21: This paragraph should be part of section 2.1, as it only refers to the atmospheric modeling. Also, it would help to have a short introduction at the very beginning of the methods section, in which the different methods that are combined (atmospheric models, marine model, element tracing) are briefly mentioned.

Throughout the manuscript (incl. figure labels), use acronyms/abbreviations consistently, i.e. do not use the same acronym for two different things (e.g. CMAQ04 for the atmospheric model and the setup of the marine model) and do not change between upper- and lowercase letters (e.g. emep and EMEP, dmu547 and DMU547)

Page 6, lines 2-4: The authors later state that EMEP data with 0.1° x 0.1° resolution are available (Karl et al., in prep. same spec. issue b). Some of the key findings of Bian et al. (2017) regarding the quality of EMEP could be included here.

Page 6, line 12 – page 7, line 16: It might make sense to first present the marine model (i.e. before the atmospheric models), as it constitutes the basis for all aspects of the study. What boundary conditions were used for the outermost model domain?

Figure 2: Are nitrification and (benthic) denitrification not included in the model? N2 fixation is also not included in the figure. I suggest including arrows for all processes that are involved in the N cycle. Also, include the description of the abbreviations in the figure caption.

Page 7, lines 13-17: It would be helpful for readers not knowing the element tracing method to at least add a short sentence how the tracing is technically done, i.e. by introducing additional model state variables and corresponding processes to the model.

Figures 3 and 4: I would suggest to merge these figures into one (my preferred choice), or at least show them as panels A and B in one figure.

Figures 5 and 7-10: The panels of the figures are too small and many features discussed in-text are barely visible.
What does "surface" in Figures 7, 9 and 10 refer to? Is it the uppermost model layer or is it a surface layer of defined thickness? Please specify in the text.
Does Figure 8 also show surface values? Please specify in the caption.
You could add the 5% and 10% isolines to the panels in the right column of Figure 9 for an easier link to the text description.

Page 13, lines 2-3: The simulated fall increase in DIN seems fine to me, considering the data variability in October/November.

Page 14, lines 29-31: What exactly do you mean with bioavailable PON? Is it detrital N only? Please add how exactly PON is calculated. Also mention that DIP is shown in the bottom row of Figure 7.

Page 17, lines 6-10: I would suggest describing the differences between the two CMAQ cases and between CMAQ and EMEP, before summarizing the section. At the moment, the latter is

not addressed and only mentioned in the last sentence of the paragraph, without any text basis for the reader.

Page 17, lines 11-16: So, we cannot draw any clear conclusion here. See my first major point

Page 17, lines 29-31: What about the effect of sinking of PON out of the surface layer? Also see my major point #2.

Page 18, line 13 – page 19, line 1: "Evaluating the total atmospheric contribution to DIN might have probably provided a qualitatively similar picture than this shipping evaluation." I do not see the relevance of this statement – maybe delete it?

Page 19, line 3: The Conclusions are too long and should rather be called Discussion. It is currently not possible to get, what the main conclusions are. The authors could further rename their Outlook to "Conclusions and Outlook" and state their most important conclusions there, followed by the suggestions for future studies.

Page 20, lines 17-18: What are the potential causes for this mismatch in the bloom timing? Please specify in the text.

Page 20, lines 28-29: This statement is very general and its validity depends strongly on the purpose and time scale of the simulation. I would argue that the high-resolution data, which resolve short-term precipitation events (see page 18, lines 4-6), are better suited for the impact analysis of such events on the local biogeochemistry.
In fact, it could be very interesting to compare station time series of, e.g. DIN and PP, produced by the different model setups at the location and time of this precipitation event, to analyze the effect of the improved resolution on local biogeochemistry.

Page 21, lines 4-5: Please explain why the relative differences in DIN and PON are much lower than those in the deposition, despite the high contribution of atmospheric deposition to nitrate at station DB2 (Figure S1)? Is the atmospheric contribution much lower in most other regions?

Page 21, lines 6-12: Please be specific what processes/interactions may not be fully covered by the model, and what "not fully covered" means in this context? Are they not included in the model or "just" under-/overestimated. Also provide references if the latter is the case.
What exactly do you mean with "improving the quality"? Is it the resolution or the magnitude of the data (considering the strong differences between CMAQ and EMEP deposition)?

Page 21, lines 19-20: The evaluation of the different deposition data is not really the goal of this study. However, considering that N deposition (from land sources) is largest in the coastal regions, I partly disagree with the statement of the authors that regions with less coastline should be considered to evaluate CMAQ and EMEP deposition data. Reliable data are especially needed in the regions with the largest atmospheric influence. These can be offshore regions as shown, e.g. for the southern North Sea (Troost et al., 2013; Große et al., 2017). However, this can also be the case in near-shore regions in the absence of major rivers (see Figure S1).

Page 21, lines 24-34: In this paragraph, the authors basically admit that the present study setup does not allow for a clear conclusion on the different study goals. From my perspective, this questions the value of the study in its current form (see also my first major point).

Page 22, lines 6-7: No actual processes are analyzed (see my second major point).

Page 22, line 15: Please state the key finding of Karl et al. (in prep. same spec. issue a).

**3. Technical corrections**

If possible within the journal manuscript guidelines, the authors could consider providing a list of abbreviations/acronyms as part of the supplement. There are a lot of these in this study, especially in the methods section. Introducing all of them in-text could be tedious.

Page 2, line 6: "StÅlnacke"

Page 2, line 18: "because a higher"

Page 3, line 24: "questions" instead of "topics"?

Page 3, lines 31-32: "to get a feeling" sounds a bit informal

Page 3, line 33: "question" instead of "topic"?

Page 4, Table 1: use uppercase acronyms (CMAQ, EMEP); use "16km x 16km" or simply "16km" for the spatial resolution (and analogous for the other resolutions; also in the text); use "Meteorology" instead of "Meteo"

Page 4, line 11: please introduce the acronym "HBM-ERGOM" here by first stating the full name

Page 5 lines 2-5: "For this study, we used CMAQ version 5.0.1 …"; remove "was used for this study"

Page 5, line 11: "online" instead of "inline"?

Page 5, line 14: "and 30 vertical z-layers each"

Page 5, line 17: There is no cmaq64 in this study

Page 5, line 33: What is "AIS"?

Page 6, line 2: "simulation is"

Page 7, line 5: "Reid1990"

Page 7, line 8: remove "Sil" in parentheses and use "SiO4-") in schematic (Fig. 2)

Page 7, line 10: add abbreviations "MiZ" and MeZ" after the two full terms

Page 7, line 15: "all model state variables and processes"

Page 8, caption Fig. 3: "Study region and geographic locations …"

Page 9, Table 2: add units for "Lon" and "Lat"; use "IOW-DB" as in the text

Page 9, line 7: "Validation data" instead of "Model validation"

Page 9, lines 9-11: remove the sentence about PON, as it is not used for validation

Page 10, lines 1-2: "less than 10km" distance from land? Please clarify. "North", "West", "South" now with uppercase first letter, previously with lowercase (page 8, line 13), please be consistent

Page 10, lines 6-11: repetition of lines 1-5

Page 10, line 14: DB2 is not yet introduced

Page 10, line 17: "because of its enclosed location and missing rivers"

Page 11, lines 5-6: This has already been mentioned on page 5, line 25

Page 11, lines 15-17: the explanation for the artifact can be shortened, as it has been done previously (page 2, lines 18-19)

Page 11, line 27: "Fig. 8", references to Figures and Tables should be in the order of their appearance in the manuscript

Page 12: line 9: What do you mean with "boxes" and "less regular patterns"?

Page 12, line 10: "Whether it"

Page 13, Figure 6: use the acronyms in the top left legend (CMAQ16 etc.)

Page 13, lines 11-12: "timing" instead of "temporal occurrence"?

Page 14, line 13: "atmospheric deposition data"

Page 14, line 22: "much lower. At station"

Page 14, line 24: "The highest differences correlate with the vicinity to land"

Page 14, line 26: "coast. Station TF13"

Page 14, line 32: add references to left and center column of Figure 7

Page 15, line 1: "Oder River"

Page 15, line 8: "patterns"; remove "except for the magnitude", it is a repetition

Page 16, Figure 8: "CMAQ04 minus CMAQ16" and "EMEP minus CMAQ16" in figure titles; add to the caption what was used for normalization to calculate the relative changes (CMAQ16?); the units of the absolute values of DIN, PON and DIP should be "μmol N/P m-3", right? Some are "μmol N/P m-2"

Page 16, line 2: "in the two columns"

Page 16, line 3: "the patterns shown for absolute differences are"

Page 17, line 10: "to EMEP nitrogen deposition"; remove "data set – namely EMEP"

Page 17, line 21: "east of Rügen"

Page 17, line 23: "Fig. 9, top right"

Page 18, line 1: "south of Funen"

Page 18, line 3: add "Lolland" label to Fig. 3

Page 18, line 5, "south of Funen"

Page 18, line 6: "due to the averaging"

Page 20, line 1: "to get a feeling" sounds very informal; "Independent of the coarse resolution, …", include reference to Karl et al. (in prep. same spec. issue b)

Page 20, line 4: "attributed" instead of "accounted for"

Page 20, lines 12-13: repetition of the previous sentence

Page 20, line 15: "was presented"

Page 20, line 17: "timing" instead of "temporal occurrence"?

Page 20, line 33: "closer" or remove "than DIN" in next line

Page 21, line 2: "therefore" instead of "wherefore"; "In the open Baltic Sea" instead of "At the open ocean"

Page 21, line 7: "fjords, boddens, and lagoons"

Page 21, line 29: "this study"

Page 22, line 15: "from a NECA scenario"

Page 22, line 24: "spatio-temporal"

Page 22, lines 31-33: shorter, because it repeats the last sentence of the previous section

Page 23, line 9: "The data is available"

Page 23, line 14: "was created"

Page 23, line 15: "publish the data, because"

Page 23, lines 16, 18, 21, 22: "data is available"

Page 23, line 22: "upon request"

Page 23, line 30: "Introduction section, and to the development of the research questions"

Page 24, lines 22-23: remove ".cdo) …", looks like a copy-paste error

---

## Referee Comment (RC2) · Anonymous Referee #2 · 17 Sep 2018

Referee comment to:

*Importance of high resolution nitrogen deposition data for biogeochemical modeling in the western Baltic Sea and the contribution of the shipping sector*

**General comments:**

The manuscript (hereafter referred to as MS) attempts to answer two relevant ocean research questions. What resolution of atmospheric nitrogen deposition is necessary to properly evaluate its' impact and how high is the contribution from shipping? The main problem with the manuscript is, as the authors seem to be aware, that the method may not allow them to fully answer the first question.

The evaluation of different spatial resolution of the nitrogen deposition is based on model results from two years, one year spin up and one year to evaluate. A typical residence time for the evaluated region is not given, but the residence time for the Baltic Sea (which is given in the MS) is stated to be significantly higher than two years. It seems reasonable to assume that a one year spin-up is too short for the biogeochemical system to reach a steady state ready to be evaluated.

However, the manuscript reads in a clear, concise, and well-structured way, although a bit repetitious. The scientific approach is transparent and the methods and results are discussed in an appropriate way. The problem is that the overall experiment design is flawed, which is a shame.

**Specific comments:**

The title of the manuscript, and the abstract, represents the manuscript's content well and only some bits of information are lacking to cleary present the methods (see the technical comments below).

One of the significant pieces of information that are missing is a statement of the quality of the physical model, HBM. This can be done by stating findings in previous publication since it is not the main focus of this MS, but it needs to be stated and the relevant publication referred to. It is not necessarily so that an operational model gives good results when used to hindcast.

Another thing missing is the lack of focus on the biogeochemical processes in the model and how they are affected by changes in the resolution of nitrogen deposition. Also, a discussion of the general nutrient dynamics of the study area should be an important part of the section "2.3 Study region". Both these additions would aid in understanding the results. Especially as the validation shows large discrepancies.

Since the study uses a constant phosphorous deposition, and aims evaluate the effect of nitrogen deposition, I assume/hope that the area is generally nitrogen limited? However, this should be stated.

The validation section does not really state that the model results are good enough to answer the research question. Are they? Is there to little data? The validation section should have a clear and well-argued conclusion.

The authors also leave too much important information to be available in other publications/submissions only. The reader should be given the most important info for this MS in a sentence or two and not only the statement that it is elsewhere. See more specific comments about this in the technical corrections section.

The conclusions section is too long and contains too much discussion and outlook.

Finally I'm skeptical to be a given a residence time valid for the Baltic Sea. Is it applicable also for the shallow Belt Sea, the southern Kattegat ect. ? If so, please state it.

**Technical corrections:**

Page 2. Line 6: Change the "Å" to lower case.

Page 2, line 21: Remove "the" before "land".

Page 2, line 31: Change "it" to "it's" or "it is".

Page 2, line 31: "Example *of* a region", not "for".

Page 3, line 17: The sentence feels unfinished.

Page 4, line 1: Consider removing the commas around "this is the first study". At least the second comma seems out of place.

Page 4, line 13-14: "Based on .... is performed". The formulation is out of place. Please, consider rewriting it.

Page 5, line 26: Please add the relevant information from the validation. I.e. are the cmaq04 simulation results sufficiently good? Are there any shortcoming the reader needs to know about?

Page 6, line 3-4: And what do Brian et al (2017) say? Since the EMEP model results are used in this MS you should shortly mention the essence of the comparison. If it is of no importance, why mention it at all?

Page 6, line 16-17: Does HBM work well in hindcast mode? Please state and/or refer to some info about the model skill.

Page 9, line 7: I think it would be beneficial to state in the "2.4 Model validation" section that the model validation will be done for all three runs.

Page 11, line 11: Remove "the" before "land".

Page 12, line 20: Replace "few" with "little" or any other word meaning "not much". "Few" is used for "not many".

Page 13, line 3: Replace "few" with "little" or any other word meaning "not much". "Few" is used for "not many".

Page 13, line 3-4: Since the winter DIP levels seem more correct than the winter DIN levels, I'd say that the high summer DIP concentrations are caused by too little DIN, not too much DIP, or possibly that there are cyanobacteria in reality, but not in the model. Something similar is stated a few lines down, but the argument is sort of started in the wrong way. Please rewrite.

Page 14, line 3: Replace "few" with "little" or any other word meaning "not much". "Few" is used for "not many".

Page 14, line 31: It is unclear if PON includes living biomass, i.e. phyto and zooplankton, or only dead particulate organic nitrogen. Does it? I suppose living matter is bio available, but it reads strange.

Page 15, line 9-10: The sentence: " This is reasonable ... and vertical stratification" makes me think PON includes living organic particulates, otherwise it makes no sense?

Page 17, line 13-15: I'm skeptical to be a given only a residence time valid for the Baltic Sea. Is it applicable also for the shallow Belt Sea, the southern Kattegat ect. ?

Page 17, line 30-31: Why is the short term impact lower?

Page 18, line 8: Mai=May ?

Page 19, fig 10: Figure it too small.

Page 20, line 8-12: But what did those publications find? Even if the evaluation work in itself is not part of this study the result is certainly important. Please use a few sentences to state the main findings instead of using two to just say that the information is elsewhere. E.g. :

"Karl et al (in prep ... issue a) found that ..... and the same study also showed that.... "

Page 20, line 12-13: Repeated sentence.

Page 20, line 28-29: Isn't EMEP better/less bad (Fig 6)?

Page 20, line 30: Why do you use a time period (April - September) when the DIN levels are likely to be depleted and thus depend to a large degree on the biogeochemical model, not so much the input? Wouldn't it be better to use winter months for DIN? If it is changes in phytoplankton growth/production you are after I suggest you show assimilation by the phytoplankton or maybe the export production or such?

---

## Author Comment (AC1) · 30 Nov 2018

**Response to the comments of Reviewer #1**

We thank the reviewer for the constructive comments on the manuscript.

**This study presents an analysis of the impact of atmospheric nitrogen (N) deposition, and in particular the contribution from ship-borne emissions, to the biogeochemistry in the western Baltic Sea. This region is affected strongly by**

atmospheric N deposition from land sources as well as from ships in the North Sea and Baltic Sea.

The authors use three different datasets (high to low spatial resolution) provided by two different atmospheric models (CMAQ and EMEP) to force their physical-biogeochemical model (HBM-ERGOM) to investigate the impact of different resolution and different data sources on the marine environment. They furthermore apply an element tracing method to the atmospheric inputs from the shipping and non-shipping sectors to quantify the contribution of the former to biogeochemical processes in the study region. A 2-year simulation (incl. a 1-year spin-up) is conducted for each of the different deposition datasets.

The authors find that the effect of different data products (i.e. atmospheric model outputs) is much larger than the effect of high vs. low resolution of the atmospheric deposition data. In addition, the contribution to the shipping sector to dissolved inorganic N (DIN) exceeds $5\%$ in large parts of the study regions, with maximum values up to $10\%$. The contribution to particulate organic N (PON) is found to be slightly lower. Hence, the shipping sector plays a small, nevertheless relevant role for the biogeochemistry in the western Baltic Sea.

The manuscript is generally well written and easy to read. However, I think in its current form the manuscript lacks a strong conclusion/scientific finding owed to the study setup. In addition, some of the analyses could be more explicitly address the research questions defined by the authors.

My strongest criticism - as also mentioned by the authors themselves - is that the current study setup, in particular the comparably short simulation period, is likely not sufficient to draw general conclusions on the impacts of both model

**resolution and the shipping sector on the biogeochemistry in the study region. Instead of referring to future studies, the authors should address at least one of the aspects they outline in their introduction to a sufficient level in order to provide a strong study and conclusion. Besides this, I have a few minor points, which I would like to see addressed before considering the manuscript for publication (outlined in the comments below).**

**Therefore, I recommend reconsidering the manuscript for publication after major revision.**

**1   Major points/General comments**

**1.1   Study setup and findings**

**From my perspective, the current study does not allow for a strong conclusion on any of the points raised by the authors (model resolution, different datasets, shipping sector), which leads to Conclusions/Outlook, which rather read as a discussion of the limitations of the setup and a referral to future studies to overcome these. As discussed by the authors, this is due to the simulation period of only two years relative to the long residence times of N (several years) in the Baltic Sea. The former is owed to the fact that the applied marine model is not designed for multi-year simulations, as it is not sufficiently parallelized. Consequently, the model is not run to steady state.**

> We extended the simulation period from two to four years for the CMAQ16mod setup and tagged the total atmospheric nitrogen deposition (not only the shipping related nitrogen deposition). A plot showing resulting DIN and chlorophyll-a concentrations is

added to the Supplementary Material (Fig. S.3). The DIN and chlorophyll-a concentrations with nitrogen of atmospheric origin (nitrogen from fixation by cyanobacteria is excluded) converge to a steady-state in the thrid year at most stations in the region of interest. The concentrations in the Arkona Basin probably need another year to converge. Additionally, the concentrations in the second year are already quite close to the third year's concentrations at most stations. Therefore, we, now, consider the two-year simulation period to be sufficient with respect to the proximity of tagged tracer concentrations to a steady-state. Nevertheless, we do not cover the interannual variablity, which would be captured by a simulation of five or ten years duration.

**I understand this technical constraint. However, there needs to be (at least) one strong and important finding that is sufficiently supported by the present study. This could be either the importance of resolution vs. different datasets or the impact of the shipping sector. The former might be easier to do, as there is already a strong indication that the different datasets overrule the effect of the different resolutions. However, it would be worthwhile to conduct this analysis based on a simulation that reached steady state, and thus provide not only qualitative indications but also a quantitative assessment of the impact. This would make the study much more relevant than it currently is.**

**Providing a comparable analysis (i.e. for a steady state) for the shipping sector's contribution would, of course, be very interesting as well, and even more relevant from a scientific point of view. However, I suppose it is computationally more expensive due to the element tagging, and the authors may want to choose only one of the two topics.**

**If neither of the two is feasible, the authors should state clearly in the introduction and conclusions that this study is only a proof of concept and more comprehensive analyses are required and left to future studies. However, this would**

**significantly reduce the impact of the study. Hence, I recommend addressing one of the two topics as described above.**

> We realized that a two-years period is not too short as we expected previously (see previous comment). We hope that this fact is sufficient as reply for this comment.

**I further wonder why the authors chose the coarse ($50 \times 50$ km$^2$) resolution EMEP dataset although there is finer resolution ($0.1° \times 0.1°$) EMEP data available? The resolution of the latter would be much closer to the CMAQ16 case, and thus would allow for a better analysis of the effects of resolution vs. different datasets. The results may not change significantly (Karl et al., in prep. same spec. issue b), nevertheless using this data also in the present study, would make it more consistent with the study goal.**

> We used the default EMEP nitrogen deposition products. For the year 2012, only data of the old resolution of $50 \times 50$ km$^2$ are available on the EMEP homepage. The year 2013 was repeated in the new resolution of $0.1° \times 0.1°$. The EMEP deposition data used in Karl et al. (in prep) were specifically generated for that publication.

1.2   Influence on biogeochemical processes

**My second major point is that the study claims to analyze the impact of resolution, datasets and shipping sector on the "biomass production" (Introduction, page 3, first research question) or "biogeochemical processes" (Conclusions, page 22, lines 6-7). However, they do not show results of actual processes, but only DIN and PON (and DIP). Instead of presenting results for PON, an analysis of the effects/contributions to primary production (PP) would be much more insightful and would explicitly address the formulated research question. In this**

**context, I would also suggest to not only analyze surface values (like in Fig. 9), but to consider water column averages (or water column integrals for PP). The differences between the shipping contributions to DIN and PON might simply be related to the sinking of (detritus as a part of) PON into deeper layers.**

> We agree that "*biomass production*" indicates that processes are consindered – which is not the case. Hence, we replaced "*biomass production*" by "*PON concentrations*".

> We also agree that looking into primary production or other process would be quite interesting. However, we did not write out the process rates during the model simulations. We would need to re-run the model to obtain *correct* process rates. We would prefer not to perform further simulations. Hence, we reformulated the research question.

**2   Minor points/Specific comments**

**Page 3, line 27: What is "this"?  Biomass production?  Please specify in the question.**

> We replaced *this* by *DIN concentrations and biomass production*.

**Page 3, lines 28-34: It should be mentioned that a marine model is used in combination with different atmospheric deposition data (question 1) and element tracing (question 2).**

> Clearly stated what is done to deal with question 1 ("*For question one, . . .*", l.28) and question 2 ("*For question two, . . .*", l. 33). Additionally in l.28–32, the sentence

"*Two nitrogen deposition data sets, which were calculated with the same CTM but with different spatial resolution and different precipitation parameterizations, are used for this purpose.*" was reformulated to "*Two nitrogen deposition data sets, which were calculated with the same CTM but with different spatial resolution and different precipitation parameterizations, are chosen for this purpose. They are used as forcing for two marine biogeochemical model simulations with one and the same model.* ".

**Although, there is a well outlined motivation for this study, it gets a bit lost in the last paragraph of the introduction (page 4, lines 2-9), which reads like a rather long summary of existing research related to the present study. The authors could simply move the first sentence (page 4, lines 1-2) to the end of the paragraph to finish the introduction with a highlight statement.**

> reordered as suggested.

**Page 4, lines 11-21: This paragraph should be part of section 2.1, as it only refers to the atmospheric modeling. Also, it would help to have a short introduction at the very beginning of the methods section, in which the different methods that are combined (atmospheric models, marine model, element tracing) are briefly mentioned.**

> The intention of this paragraph (p.4, l.11–21) was to give such quick broad description of the model setup here before describing the model components and input data in detail. We modified the text passages in the beginning of section 2 in order to better fulfill this.

**Throughout the manuscript (incl. figure labels), use acronyms/abbreviations consistently, i.e. do not use the same acronym for two different things (e.g.**

**CMAQ04 for the atmospheric model and the setup of the marine model) and do not change between upper- and lowercase letters (e.g. emep and EMEP, dmu547 and DMU547)**

> All station names are written upper case, now.

> Deposition data are denoted as CMAQ04dep, CMAQ16dep, and EMEP50dep.

> Marine biogeochemical model results are denoted as CMAQ04mod, CMAQ16mod, and EMEP50mod.

> All figures and tables were updated to comply with the new naming convention.

**Page 6, lines 2-4: The authors later state that EMEP data with $0.1° \times 0.1°$ resolution are available (Karl et al., in prep. same spec. issue b). Some of the key findings of Bian et al. (2017) regarding the quality of EMEP could be included here.**

> a) We used the default EMEP nitrogen deposition products. For the year 2012, only data of the old resolution of $50 \times 50$ km$^2$ are available on the EMEP homepage. The year 2013 was repeated in the new resolution of $0.1° \times 0.1°$. The EMEP deposition data used in Karl et al. (in prep) were specifically generated for that publication.

> b) We included a paragraph summarizing the results of Bian et al. (2017).

**Page 6, line 12 - page 7, line 16: It might make sense to first present the marine model (i.e. before the atmospheric models), as it constitutes the basis for all aspects of the study. What boundary conditions were used for the outermost model domain?**

> We have chosen the order of atmospheric (Sect. 2.1) and oceanic (Sect. 2.2) model description because this reflects the order of the nitrogen flow considered in this study

(going from the atmosphere into the ocean). Therefore, we think that the present order is reasonable.

> Appended information on boundary conditions to the paragraph in grid resolution (p. 7, l. 21)

**Figure 2: Are nitrification and (benthic) denitrification not included in the model? N2 fixation is also not included in the figure. I suggest including arrows for all processes that are involved in the N cycle. Also, include the description of the abbreviations in the figure caption.**

> nitrification, denitrification, and fixation are now included as suggested; a second figure showing processes in the sediment and at the sediment-water interface was added;

> added all abbreviations to all figure captions;

**Page 7, lines 13-17: It would be helpful for readers not knowing the element tracing method to at least add a short sentence how the tracing is technically done, i.e. by introducing additional model state variables and corresponding processes to the model.**

> inserted as a new paragraph after line 17

**Figures 3 and 4: I would suggest to merge these figures into one (my preferred choice), or at least show them as panels A and B in one figure.**

> Figures merged as suggested

**Figures 5 and 7-10: The panels of the figures are too small and many features discussed in-text are barely visible.**

> Accidently, we created the submitted PDF with medium resolution PNG files instead of vector graphics. Now we inserted them as vector graphics and increased their width in the document.

**What does "surface" in Figures 7, 9 and 10 refer to? Is it the uppermost model layer or is it a surface layer of defined thickness? Please specify in the text.**

> Included: "*The upper five model layers (*$0$ m *to* $\approx 12$ m *depth) are considered as surface layer.*". We included this information also in the first paragraph of Sect. 2.4 ("*Model Validation*").

**Does Figure 8 also show surface values? Please specify in the caption.**

> Yes, surface values. Added to captions of Figs. 7 and 8.

**You could add the 5% and 10% isolines to the panels in the right column of Figure 9 for an easier link to the text description.**

> isolines added as suggested

**Page 13, lines 2-3: The simulated fall increase in DIN seems fine to me, considering the data variability in October/November.**

> Removed the sentence.

**Page 14, lines 29-31: What exactly do you mean with bioavailable PON? Is it detrital N only? Please add how exactly PON is calculated. Also mention that DIP is shown in the bottom row of Figure 7.**

> We explicitly state "bioavailable" because we do not consider PON that is somehow not available for biogeochemical processes. Clarified the meaning of "*bioavailable PON*", added how PON is calculated (detritus + phytoplankton + zooplankton), and mentioned DIP.

**Page 17, lines 6-10: I would suggest describing the differences between the two CMAQ cases and between CMAQ and EMEP, before summarizing the section. At the moment, the latter is not addressed and only mentioned in the last sentence of the paragraph, without any text basis for the reader.**

> added another sentence to the CMAQ04-CMAQ16 comparison and added a CMAQ-EMEP comparison as suggested

**Page 17, lines 11-16: So, we cannot draw any clear conclusion here. See my first major point**

> split the paragraph into two and modified the content; simulations over longer time periods indicate that a steady-state of atmospheric nitrogen is reach after two to three years;

**Page 17, lines 29-31: What about the effect of sinking of PON out of the surface layer? Also see my major point #2.**

> We ignore the PON, which leaves the surface layer. We agree that it would be very important for budget calculations. Instead of dealing with this issue we reformulated the research questions by prepending "*surface layer*" to "*concentrations*".

**Page 18, line 13 - page 19, line 1: "Evaluating the total atmospheric contribution to DIN might have probably provided a qualitatively similar picture than this shipping evaluation." I do not see the relevance of this statement - maybe delete it?**

> removed sentence

**Page 19, line 3: The Conclusions are too long and should rather be called Discussion. It is currently not possible to get, what the main conclusions are. The authors could further rename their Outlook to "Conclusions and Outlook" and state their most important conclusions there, followed by the suggestions for future studies.**

> We renamed the *conclusions* to *summarizing discussion*, included the suggested modifications from below, and wrote new *conclusions*.

**Page 20, lines 17-18: What are the potential causes for this mismatch in the bloom timing? Please specify in the text.**

> We added Fig. S.1 to the Supplement (previous Fig. S.1 now is Fig. S.2a), which shows the modeled and measured salinity in the Arkona Basin between 1st and 5th February 2012. It shows that the modeled Baltic Sea was stratified (blue line) whereas the real Baltic Sea (black "+") was well mixed. Therefore, the modeled algal bloom

started in February, while the real-world algal bloom ddid not start before March (Was-mund et al., 2013).

> Similarly, in the end of the year, the mixing starts too early in the model.

> Added the information to the paragraph p.13 l.11 to p.14 l.4 (old page and line numbers). Added a brief statement to p.20, l.18.

**Page 20, lines 28-29: This statement is very general and its validity depends strongly on the purpose and time scale of the simulation. I would argue that the high-resolution data, which resolve short-term precipitation events (see page 18, lines 4-6), are better suited for the impact analysis of such events on the local biogeochemistry. In fact, it could be very interesting to compare station time series of, e.g. DIN and PP, produced by the different model setups at the location and time of this precipitation event, to analyze the effect of the improved resolution on local biogeochemistry.**

> removed the sentence as suggested and re-wrote similar as suggested for new conclusions

**Page 21, lines 4-5: Please explain why the relative differences in DIN and PON are much lower than those in the deposition, despite the high contribution of atmospheric deposition to nitrate at station DB2 (Figure S1)? Is the atmospheric contribution much lower in most other regions?**

> Note (because Supplement changed): previous Fig. S.1 now is S.2b.

> The atmospheric contribution is much lower in most other regions. We added a new figure Fig. S.5 to the supplement showing relative atmospheric contribution.

> The sentences was ambiguously formulated. It should refer to the open sea and not to the Grabow. We inserted a paragraph break on p.21 l.2 to emphasize the geographic shift and reformulated "*However, the relative differences in the DIN and PON concentrations were considerably lower than the relative differences in the nitrogen deposition.*" to "*However in this study, the relative differences between CMAQ16mod and CMAQ04mod in the DIN and PON concentrations were considerably lower than the relative differences between CMAQ16dep and CMAQ04dep.*". Additionally, we appende the sentences "*This is reasonable because nitrogen in DIN does not only source from atmospheric deposition. Hence, a $10\%$ deviation in the nitrogen deposition is less than $10\%$ deviation in DIN.*".

> Nevertheless, the situation at DB2 is quite interesting. The dominant source for DIN in the Grabow is atmospheric nitrogen deposition. Relative differences between $\mathrm{DIN}_{CMAQ16mod}$ and $\mathrm{DIN}_{CMAQ04mod}$ and between $\mathrm{PON}_{CMAQ16mod}$ and $\mathrm{PON}_{CMAQ04mod}$ should be close to relative differences between CMAQ16dep and CMAQ04dep. For DIN, this is the case. For PON, this is not the case (Fig. 8, left column; Grabow is not red for PON). We added another Figure to the Supplement (Fig. S.2b), which is similar to Fig. S.2a but shows PON instead of DIN. We see that the relative atmospheric contribution to PON (absolute values are shown!) is lower than the relative atmospheric contribution to DIN. The PON concentrations show similar oscillations as the DIP concentrations (Fig. 6, column four, center row). Hence, we assume that PON from outside the Grabow is transported ocasionally into the lagoon. DIN is also transported into the Grabow. But, DIN is quickly consumed by phytoplankton, whereas PON has a longer residence time. Therefore, the non-atmospheric nitrogen contribution to PON is higher than to DIN.

**Page 21, lines 6-12: Please be specific what processes/interactions may not be fully covered by the model, and what "not fully covered" means in this context? Are they not included in the model or "just" under-/overestimated. Also provide references if the latter is the case. What exactly do you mean with "improving**

**the quality"? Is it the resolution or the magnitude of the data (considering the strong differences between CMAQ and EMEP deposition)?**

> removed the sentences containing "*improving the quality*"

> re-wrote sentence to "*The reasons for this might be (a) missing small rivers and diffusive ground water inflows and (b) simplified water-sediment interactions (e.g. compare this study's sediment processes with Asmala et al. (2017))*."

**Page 21, lines 19-20: The evaluation of the different deposition data is not really the goal of this study. However, considering that N deposition (from land sources) is largest in the coastal regions, I partly disagree with the statement of the authors that regions with less coastline should be considered to evaluate CMAQ and EMEP deposition data. Reliable data are especially needed in the regions with the largest atmospheric influence. These can be offshore regions as shown, e.g. for the southern North Sea (Troost et al., 2013; Große et al., 2017). However, this can also be the case in near-shore regions in the absence of major rivers (see Figure S1).**

> We removed the sentence.

> We think that the models should be valided on the plain ocean first because there we have less processes and side effects to consider. If we are sure that the nitrogen deposition into the ocean is correctly calculated we can go to more complicated areas like coastal reagions.

**Page 21, lines 24-34: In this paragraph, the authors basically admit that the present study setup does not allow for a clear conclusion on the different study goals. From my perspective, this questions the value of the study in its current form (see also my first major point).**

> removed large parts of this paragraph and rewrote the conclusions completely

**Page 22, lines 6-7: No actual processes are analyzed (see my second major point).**

> "*processes*" was removed from the research questions

**Page 22, line 15: Please state the key finding of Karl et al. (in prep. same spec. issue a).**

> Removed sentence because extending this topic would take some space and provide no benefit for the conclusions of this study.

**3 Technical corrections**

**If possible within the journal manuscript guidelines, the authors could consider providing a list of abbreviations/acronyms as part of the supplement. There are a lot of these in this study, especially in the methods section. Introducing all of them in-text could be tedious.**

> added a list of abbreviations/acronyms to the supplement as suggested

**Page 2, line 6: "StÅlnacke"**

> corrected as suggested

**Page 2, line 18: "because a higher"**

> corrected as suggested

**Page 3, line 24: "questions" instead of "topics"?**

> changed as suggested

**Page 3, lines 31-32: "to get a feeling" sounds a bit informal**

> changed *"to get a feeling"* to *"to assess"*

**Page 3, line 33: "question" instead of "topic"?**

> changed as suggested

**Page 4, Table 1: use uppercase acronyms (CMAQ, EMEP); use "16km x 16km" or simply "16km" for the spatial resolution (and analogous for the other resolutions; also in the text); use "Meteorology" instead of "Meteo"**

> Modified structure of the table. Renamed deposition data sets to: CMAQ16dep, CMAQ04dep, and EMEP50dep. The model results of the simulations forced by these deposition data sets are now denoted as CMAQ16mod, CMAQ04mod, and EMEP50mod, respectively. Because of the new naming convention the reader directly knows if a deposition data set or biogeochemical model result is mentioned in the text.

**Page 4, line 11: please introduce the acronym "HBM-ERGOM" here by first stating the full name**

> changed *"HBM-ERGOM"* to *"HBM-ERGOM (HIROMB-BOOS-Model, Ecological Re-Gional Ocean Model)"*

**Page 5 lines 2-5: "For this study, we used CMAQ version 5.0.1 . . . "; remove "was used for this study"**

> changed as suggested

**Page 5, line 11: "online" instead of "inline"?**

> changed as suggested

**Page 5, line 14: "and 30 vertical z-layers each"**

> changed as suggested

**Page 5, line 17: There is no cmaq64 in this study**

> removed as suggested

**Page 5, line 33: What is "AIS"?**

> Included on page 5 in former line 10 after *"Marine shipping emissions were calcu-lated with the STEAM mode (Jalkanen et al., 2012)"*. Now it reads *"Marine shipping*

*emissions were calculated with the STEAM model (Jalkanen et al., 2012) based on data of the automatic identification system (AIS). Via AIS modern ships broadcast their location, direction of travel, speed, IMO number, and further information."*

**Page 6, line 2: "simulation is"**

> Modified whole sentence.

**Page 7, line 5: "Reid1990"**

> corrected as suggested

**Page 7, line 8: remove "Sil" in parentheses and use "SiO4-" in schematic (Fig. 2)**

> corrected in text and figure as suggested

**Page 7, line 10: add abbreviations "MiZ" and "MeZ" after the two full terms**

> added suggested

**Page 7, line 15: "all model state variables and processes"**

> replaced *"the ecosystem"* by the new formulation

**Page 8, caption Fig. 3: "Study region and geographic locations . . . "**

> added as suggested

**Page 9, Table 2: add units for "Lon" and "Lat"; use "IOW-DB" as in the text**

> Added units [°E] and [°N]. Corrected *"IOW"* to *"IOWDB"*.

**Page 9, line 7: "Validation data" instead of "Model validation"**

> corrected as suggested

**Page 9, lines 9-11: remove the sentence about PON, as it is not used for validation**

> removed as suggested

**Page 10, lines 1-2: "less than 10km" distance from land? Please clarify. "North", "West", "South" now with uppercase first letter, previously with lowercase (page 8, line 13), please be consistent**

> Replaced *". . . and is surrounded by land (less than 10 km) towards North, West, and South."* by *". . . . Its distance to land towards north, west, and south is less than* 10 km *each.".*

**Page 10, lines 6-11: repetition of lines 1-5**

> removed as suggested

**Page 10, line 14: DB2 is not yet introduced**

> Switched order of TF12 and DB2 (now: first DB2, then TF12)

**Page 10, line 17: "because of its enclosed location and missing rivers"**

> added as suggested

**Page 11, lines 5-6: This has already been mentioned on page 5, line 25**

> removed as suggested

**Page 11, lines 15-17: the explanation for the artifact can be shortened, as it has been done previously (page 2, lines 18-19)**

> removed some parts and reformulated text passage a bit; In the Introduction (p.2, l.18-19), we mention that the dry deposition is higher above land than above water. However, we do not describe, which artefacts in the model data result from this when common output strategies (sum deposition into/onto all compartments) are used.

**Page 11, line 27: "Fig. 8", references to Figures and Tables should be in the order of their appearance in the manuscript**

> added to maps from Fig. 8 to Fig. 5

**Page 12: line 9: What do you mean with "boxes" and "less regular patterns"?**

[Figure]

> Replaced *"regular boxes but a less regular pattern"* by *"squares of* 50 km *side length"*

**Page 12, line 10: "Whether it"**

> added as suggested

**Page 13, Figure 6: use the acronyms in the top left legend (CMAQ16 etc.)**

> acronyms added; some minor font improvements

**Page 13, lines 11-12: "timing" instead of "temporal occurrence"?**

> replaced as suggested

**Page 14, line 13: "atmospheric deposition data"**

> added *"data"* as suggested

**Page 14, line 22: "much lower. At station"**

> corrected as suggested

**Page 14, line 24: "The highest differences correlate with the vicinity to land"**

> corrected as suggested

**Page 14, line 26: "coast. Station TF13"**

> corrected as suggested

**Page 14, line 32: add references to left and center column of Figure 7**

> corrected as suggested

**Page 15, line 1: "Oder River"**

> corrected as suggested

**Page 15, line 8: "patterns"; remove "except for the magnitude", it is a repetition**

> modified and removed as suggested

**Page 16, Figure 8: "CMAQ04 minus CMAQ16" and "EMEP minus CMAQ16" in figure titles; add to the caption what was used for normalization to calculate the relative changes (CMAQ16?); the units of the absolute values of DIN, PON and DIP should be "µmol N/P m$^{-3}$", right? Some are "µmol N/P m$^{-2}$"**

> Added title to figures. We replaced *"with respect"* by *"relative"*. We corrected the units (right, it should be $m^{-3}$).)

**Page 16, line 2: "in the two columns"**

> added *"the"* as suggested
**Page 16, line 3: "the patterns shown for absolute differences are"**

> corrected as suggested

**Page 17, line 10: "to EMEP nitrogen deposition"; remove "data set - namely EMEP"**

> Replaced *"to another nitrogen deposition data set - namely EMEP"* by *"to the EMEP nitrogen deposition data set"*

**Page 17, line 21: "east of R$^\prime$ugen"**

> corrected as suggested

**Page 17, line 23: "Fig. 9, top right"**

> corrected as suggested

**Page 18, line 1: "south of Funen"**

> corrected as suggested

**Page 18, line 3: add "Lolland" label to Fig. 3**

> updated as suggested

**Page 18, line 5, "south of Funen"**

> corrected as suggested

**Page 18, line 6: "due to the averaging"**

> corrected as suggested

**Page 20, line 1: "to get a feeling" sounds very informal; "Independent of the coarse resolution, . . . ", include reference to Karl et al. (in prep. same spec. issue b)**

> Replaced *"to get a feeling"* by *"to assess"*. Changed *"coarser"* to *"coarse"*. A reference to Karl et al. is included in line 7 and onwards. Line 1 describes differences that are visible in this publication's figures. Therefore, we did not add a reference to Karl et al. there.

**Page 20, line 4: "attributed" instead of "accounted for"**

> corrected as suggested

**Page 20, lines 12-13: repetition of the previous sentence**

> removed duplicate as suggested

**Page 20, line 15: "was presented"**

> We did include this correction: *"Data . . . were . . . ."*

**Page 20, line 17: "timing" instead of "temporal occurrence"?**

> corrected as suggested

**Page 20, line 33: "closer" or remove "than DIN" in next line**

> corrected as suggested

**Page 21, line 2: "therefore" instead of "wherefore"; "In the open Baltic Sea" instead of "At the open ocean"**

> Replaced *"wherefore"* by *"therefore"* and split this sentence into two by adding a full stop before *"therefore"*. Included other correction.

**Page 21, line 7: "fjords, boddens, and lagoons"**

> corrected as suggested

**Page 21, line 29: "this study"**

> corrected as suggested

**Page 22, line 15: "from a NECA scenario"**

> corrected as suggested

**Page 22, line 24: "spatio-temporal"**

> corrected as suggested

**Page 22, lines 31-33: shorter, because it repeats the last sentence of the previous section**

> Replaced *"In this study, only one year was considered. However, the impact of different deposition data sets might take several years to develop its full impact. Therefore, future studies should focus on a longer time period – i.e. five to ten years when nitrogen is considered."* by *"Future studies should focus on time periods of five to ten years because the impact of different deposition data sets might take several years to develop its full impact."*.

**Page 23, line 9: "The data is available"**

> Not corrected. The word *data* is considered to be plural.

**Page 23, line 14: "was created"**

> Not corrected. See above.

**Page 23, line 15: "publish the data, because"**

> corrected as suggested

**Page 23, lines 16, 18, 21, 22: "data is available"**

> Not corrected. See above.

**Page 23, line 22: "upon request"**

> corrected as suggested

**Page 23, line 30: "Introduction section, and to the development of the research questions"**

> corrected as suggested

**Page 24, lines 22-23: remove ".cdo) . . . ", looks like a copy-paste error**

> removed as suggested

---

## Author Comment (AC2) · 30 Nov 2018

**Response to the comments of Reviewer #2**

We thank the reviewer for the constructive comments on the manuscript.

[Figure]

**1 General comments**

The manuscript (hereafter referred to as MS) attempts to answer two relevant ocean research questions. What resolution of atmospheric nitrogen deposition is necessary to properly evaluate its' impact and how high is the contribution from shipping? The main problem with the manuscript is, as the authors seem to be aware, that the method may not allow them to fully answer the first question.

The evaluation of different spatial resolution of the nitrogen deposition is based on model results from two years, one year spin up and one year to evaluate. A typical residence time for the evaluated region is not given, but the residence time for the Baltic Sea (which is given in the MS) is stated to be significantly higher than two years. It seems reasonable to assume that a one year spin-up is too short for the biogeochemical system to reach a steady state ready to be evaluated.

However, the manuscript reads in a clear, concise, and well-structured way, although a bit repetitious. The scientific approach is transparent and the methods and results are discussed in an appropriate way. The problem is that the overall experiment design is flawed, which is a shame.

**2 Specific comments**

The title of the manuscript, and the abstract, represents the manuscript's content well and only some bits of information are lacking to cleary present the methods (see the technical comments below).

**One of the significant pieces of information that are missing is a statement of the quality of the physical model, HBM. This can be done by stating findings in previous publication since it is not the main focus of this MS, but it needs to be stated and the relevant publication referred to. It is not necessarily so that an operational model gives good results when used to hindcast.**

> We added a second paragraph to Sect. 2.2 ("*Marine Modeling*") providing information on the quality of the physical mode.

**Another thing missing is the lack of focus on the biogeochemical processes in the model and how they are affected by changes in the resolution of nitrogen deposition. Also, a discussion of the general nutrient dynamics of the study area should be an important part of the section "2.3 Study region". Both these additions would aid in understanding the results. Especially as the validation shows large discrepancies.**

> added a description of basic system dynamics to Sect. 2.3 as suggested

> removed "*processes*" from the research question

**Since the study uses a constant phosphorous deposition, and aims evaluate the effect of nitrogen deposition, I assume/hope that the area is generally nitrogen limited? However, this should be stated.**

> Yes, most parts of the Baltic Sea are nitrogen limited (Feistel et al., 2008, Sect. 12.3, Table 12.3). Nutrient loads of rivers might habe N:P ratios above 16 (Svendsen et al., 2015). However, the areas affected by river plumes and phosphorus limitation are rather small.

> added this information to Sect. 2.3 and pointed from phosphorus-deposition-paragraph in Sect. 2.2 to Sect. 2.3

> According to current assumption by HELCOM (2015b, "Updated Fifth Baltic Sea Pollution Load Compilation (PLC-5.5)"), atmospheric phosphorous deposition contributes approximately $5\,\%$ to the total phosphorous input into the Baltic Sea. Therefore, phosphorous deposition has a lower contribution than nitrogen deposition with respect to the total inputs. However, there is large uncertainty in phosphorous deposition estimates and a few colleagues expect the phosphorous deposition to be higher.

**The validation section does not really state that the model results are good enough to answer the research question. Are they? Is there too little data? The validation section should have a clear and well-argued conclusion. The authors also leave too much important information to be available in other publications/submissions only. The reader should be given the most important info for this MS in a sentence or two and not only the statement that it is elsewhere. See more specific comments about this in the technical corrections section.**

> We added a summarizing paragraph to the validation section as suggested.

> We added information from cited publications – i.e. Bian et al. (2017) and Karl et al. (2018a,b) – to the text as documented in our answers to the specific comments below.

**The conclusions section is too long and contains too much discussion and outlook.**

> We renamed the *conclusions* to *summarizing discussion*, included the suggested modifications from below, and wrote new *conclusions*.

**Finally I'm skeptical to be a given a residence time valid for the Baltic Sea. Is**

**it applicable also for the shallow Belt Sea, the southern Kattegat ect. ? If so, please state it.**

> The residence time cited from Radtke et al. (2012) should be considered as an upper limit because the region of interest is quite shallow – as the reviewer mentions. We modified the respective paragraphs in the Section 3.2.3 (last paragraph) and in the Conclusions.

**3 Technical corrections**

Page 2. Line 6: Change the "Å" to lower case.

> corrected as suggested

Page 2, line 21: Remove "the" before "land".

> removed as suggested

Page 2, line 31: Change "it" to "it's" or "it is".

> changed as suggested

Page 2, line 31: "Example of a region", not "for".

> corrected as suggested

Page 3, line 17: The sentence feels unfinished.

> We changed "*However, shipping traffic is also expected to increase in the Baltic Sea in the next decades (e.g., Matthias et al., 2016; Karl et al., in prep. same spec. issue a).*" to "*However, shipping traffic is also expected to increase in the Baltic Sea in the next decades and cargo vessels have a life time of approximately 25 to 30 years (e.g., Buhaug et al., 2009; Matthias et al., 2016; Karl et al., in prep., a; Smith et al., 2014; Danish EPA, 2012). Therefore, the expected reduction in overall shipping $NO_X$ emissions is rather low in the next decade (e.g., Geels et al., 2012; Jonson et al., 2015; Hammingh et al., 2012).*"

Page 4, line 1: Consider removing the commas around "this is the first study". At least the second comma seems out of place.

> removed commas as suggested; we restructured this paragraph and moved this sentence to the end of the paragraph;

Page 4, line 13-14: "Based on . . . is performed". The formulation is out of place. Please, consider rewriting it.

> replaced "*Based on . . . is performed*" by "*The impact of spatially refined nitrogen deposition on the biogeochemical model results was assessed by evaluating two HBM-ERGOM model simulations forced by these two deposition data sets. These HBM-ERGOM simulations are named CMAQ16mod and CMAQ04mod, respectively, corresponding to the deposition data sets.*"

Page 5, line 26: Please add the relevant information from the validation. I.e. are the cmaq04 simulation results sufficiently good? Are there any shortcoming the reader needs to know about?

> The nitrogen wet deposition was underestimated compared to EMEP measurements in the Baltic Sea region. However, the atmospheric $NO_X$ were well reproduced. The reason for the underestimation of the wet deposition was not fully resolved. We extended the paragraph.

Page 6, line 3-4: And what do Bian et al (2017) say? Since the EMEP model results are used in this MS you should shortly mention the essence of the comparison. If it is of no importance, why mention it at all?

> Thanks for the remark. We included a paragraph summarizing the results of Bian et al. (2017).

Page 6, line 16-17: Does HBM work well in hindcast mode? Please state and/or refer to some info about the model skill.

> We appended the following sentences to the paragraph: "*Sea surface temperature and salinity are generally well predicted according to recent publications (Wan et al., 2012; Bruening et al., 2014*. However, Wan et al. (2012) identified a drift of salinity in depths $> 75$ m of the central Baltic Sea in a two-year hindcast indicating too strong vertical mixing. In preliminary multi-year simulations we found similar evidence. Because we evaluate a short time span and surface water concentrations only, we expect no critical interference through this issue."

Page 9, line 7: I think it would be beneficial to state in the "2.4 Model validation" section that the model validation will be done for all three runs.

> changed sentence on p.9 l.8 from "*The biogeochemical model results were validated against measurements.*" to "*The results of all three HBM-ERGOM simulations were validated against measurements.*"

Page 11, line 11: Remove "the" before "land".

> removed as suggested

Page 12, line 20: Replace "few" with "little" or any other word meaning "not much". "Few" is used for "not many".

> corrected as suggested

Page 13, line 3: Replace "few" with "little" or any other word meaning "not much". "Few" is used for "not many".

> corrected as suggested

Page 13, line 3-4: Since the winter DIP levels seem more correct than the winter DIN levels, I'd say that the high summer DIP concentrations are caused by too little DIN, not too much DIP, or possibly that there are cyanobacteria in reality, but not in the model. Something similar is stated a few lines down, but the argument is sort of started in the wrong way. Please rewrite.

> rewritten as suggested

> added a sentences on this topic to the "Summarizing Discussion" (previously "Conclusions")

Page 14, line 3: Replace "few" with "little" or any other word meaning "not much". "Few" is used for "not many".

> corrected as suggested

Page 14, line 31: It is unclear if PON includes living biomass, i.e. phyto and zooplankton, or only dead particulate organic nitrogen. Does it? I suppose living matter is bio available, but it reads strange.

> Yes, PON as we use it includes living biomass. We explicitely state "bioavailable" because we do not consider PON that is somehow not available for biogeochemical processes. Clarified the meaning of "*bioavailable PON*" and added how PON is calculated (detritus + phytoplankton + zooplankton).

Page 15, line 9-10: The sentence: " This is reasonable . . . and vertical stratification" makes me think PON includes living organic particulates, otherwise it makes no sense?

> Yes. Please see reply to previous comment.

Page 17, line 13-15: I'm skeptical to be a given only a residence time valid for the Baltic Sea. Is it applicable also for the shallow Belt Sea, the southern Kattegat ect. ?

> We performed additional three years of simulation and saw that atmospheric nitrogen converges to a steady state after two to three years in this region. We, now, consider the numbers of Radtke et al. (2012) as upper limit. Therefore, the two-years period evaluated in this study seems to by only just sufficient.

> We split the paragraph into two and modified the text according to the new findings.

Page 17, line 30-31: Why is the short term impact lower?

> The whole paragraph was reformulated. Originally, we wanted to express that the steady-state atmospheric nitrogen concentration is not reached yet. It was badly formulated.

Page 18, line 8: Mai=May ?

> corrected as suggested

Page 19, fig 10: Figure it too small.

> increased size and included it as vector graphic (before it was a PNG)

Page 20, line 8-12: But what did those publications find? Even if the evaluation work in itself is not part of this study the result is certainly important. Please use a few sentences to state the main findings instead of using two to just say that the information is elsewhere. E.g. : "Karl et al (in prep . . . issue a) found that . . . and the same study also showed that . . . "

> reformulated second half of paragraph and moved to summary (now first paragraph of summary)

Page 20, line 12-13: Repeated sentence.

> removed as suggested

Page 20, line 28-29: Isn't EMEP better/less bad (Fig 6)?

> We agree that summer DIP concentrations and winter DIN are closer to the mea-
surements in the EMEP50mod case. A sentence was added to the new Sect. 3.5
(*Summarizing Discussion*) "*. . . , EMEP50mod DIN and DIP concentrations were closer
to the measurements in these particular time periods.*"

Page 20, line 30: Why do you use a time period (April - September) when the DIN levels
are likely to be depleted and thus depend to a large degree on the biogeochemical
model, not so much the input? Wouldn't it be better to use winter months for DIN?
If it is changes in phytoplankton growth/production you are after I suggest you show
assimilation by the phytoplankton or maybe the export production or such?

> We agree. Now, we show annual means to cover the nutrient-rich period in winter and
the phytoplankton-rich period in summer. Unfortunately, we did not write out nutrient
consumption/release per process. Hence, a detailed evaluation of net process flows is
not possible without re-running the simulations.

---

## Author Comment (AC3) · 4 Dec 2018

**Supplement: Importance of high resolution nitrogen deposition data for biogeochemical modeling in the western Baltic Sea and the contribution of the shipping sector**

Figure S.1 contains four vertical salinity profiles at the station TF113. Sub-figures (a) and (b) indicate differences between measurement and model in the timing of the onset of the stratification in the beginning of 2012. The stratification develops earlier in the model than in the reality. Sub-figures (c) and (d) indicate differences between measurement and model in the timing of the decline of the stratification in autumn. The mixing starts earlier in the model than in reality.

Figure S.1: Measured (black cross) and modeled (blue line) vertical salinity profiles at the stations TF113 during four different time periods.

**Supplement to Neumann et al., 2018: Importance of high resolution nitrogen deposition data for ...**

Figure S.2a shows the contribution of shipping-related and total atmospheric (shipping + everything else) nitrogen deposition to the nitrate  $(NO_3^-; top)$  and particulate organic nitrogen (PON; bottom) concentrations at the station DB2. These results are based on the CMAQ16mod simulation. Additionally, the total nitrate concentrations of the CMAQ16mod and CMAQ04mod simulations are plotted. One clearly sees that nitrogen from atmospheric deposition (red) dominates the nitrate concentrations (total is plotted in blue). Figure S.2b shows the same but for particulate organic nitrogen (PON).